# RESIDUAL DEEP GAUSSIAN PROCESSES ON MANIFOLDS

**Kacper Wyrwal**
ETH Zürich
University of Edinburgh

**Andreas Krause**
ETH Zürich

**Viacheslav Borovitskiy**
ETH Zürich

## ABSTRACT

We propose practical deep Gaussian process models on Riemannian manifolds, similar in spirit to residual neural networks. With manifold-to-manifold hidden layers and an arbitrary last layer, they can model manifold- and scalar-valued functions, as well as vector fields. We target data inherently supported on manifolds, which is too complex for shallow Gaussian processes thereon. For example, while the latter perform well on high-altitude wind data, they struggle with the more intricate, nonstationary patterns at low altitudes. Our models significantly improve performance in these settings, enhancing prediction quality and uncertainty calibration, and remain robust to overfitting, reverting to shallow models when additional complexity is unneeded. We further showcase our models on Bayesian optimisation problems on manifolds, using stylised examples motivated by robotics, and obtain substantial improvements in later stages of the optimisation process. Finally, we show our models to have potential for speeding up inference for non-manifold data, when, and if, it can be mapped to a proxy manifold well enough.

## 1 INTRODUCTION

Gaussian processes (GPs) are a widely adopted model class for learning functions within the Bayesian framework (Rasmussen and Williams, 2006). They offer accurate uncertainty estimates and perform well even when data is scarce. Consequently, GPs have found success in decision-making tasks, where well-calibrated uncertainty is key, including Bayesian optimisation (Snoek et al., 2012), active (Krause et al., 2008) and reinforcement (Kamthe and Deisenroth, 2018) learning.

In recent years, substantial work went into developing the analogues of practical GP models on various non-Euclidean domains (Borovitskiy et al., 2021; 2023; 2020; Fichera et al., 2023). By virtue of being geometry-aware, these analogues have demonstrated improved performance in a variety of tasks on non-Euclidean spaces. Their notable applications include Bayesian optimisation on manifolds for robotics (Jaquier et al., 2022), traffic flow interpolation on road networks (Borovitskiy et al., 2021), and wind velocity prediction on the globe (Hutchinson et al., 2021; Robert-Nicoud et al., 2024). These models were also used to speed up inference for Euclidean GPs by transferring data to a hypersphere and leveraging the attractive structure GPs thereon possess (Dutordoir et al., 2020).

Despite their advantages, GPs can sometimes fall short in modelling complex, irregular functions. To address this, deep Gaussian processes have been introduced as a sequential composition of GPs (Damianou and Lawrence, 2013), providing improved flexibility through their layered structure (Dai et al., 2016; Mattos et al., 2016). Many techniques developed for shallow GPs, such as variational inference (Salimbeni and Deisenroth, 2017) and efficient sampling techniques (Wilson et al., 2020), can be adapted for deep GPs, enabling them to be efficiently trained and deployed on large datasets. This scalability is vital when dealing with data with complex, irregular patterns.

Deep Gaussian processes have demonstrated success in handling complex data of Euclidean nature, often competing with Bayesian neural networks and deep ensembles. However, there has been limited work on expressive uncertainty-quantifying models on manifolds beyond shallow GPs. This gap leads to the natural question: how can we construct deep Gaussian processes on manifolds?

By analogy with the Euclidean case, a deep GP on a manifold should be a composition of GP layers which take inputs and produce outputs on the manifold of interest. While significant advancements have been made in handling manifold-input GPs, outputs on manifolds conflict with the fundamental

---

Correspondence to wyrwal.kacper@gmail.com and viacheslav.borovitskiy@gmail.com.
Code available at https://github.com/KacperWyrwal/residual-deep-gps.

Hidden Layer $\times L$

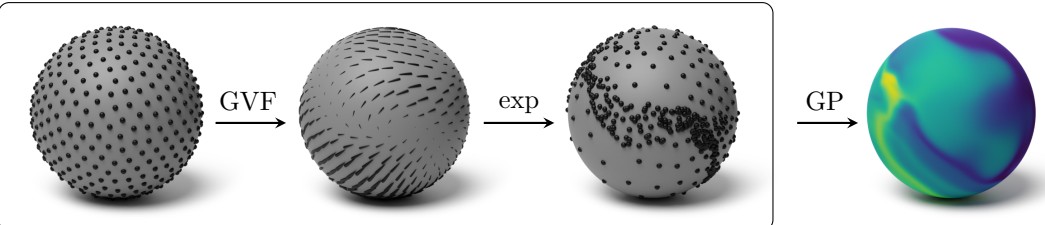

Figure 1: Schematic illustration of a scalar-valued residual deep GP with $L$ hidden layers. The last layer is a scalar-valued GP on the manifold. If it is not present, the model is manifold-valued. If it is replaced with a Gaussian vector field (GVF), the model is a vector field on the manifold.

concept of a GP, which dictates that outputs must be Gaussian and thus Euclidean.[1] Designing GPs with inputs or outputs on a manifold is challenging; with both, it is even more so.

Our solution to this problem is in part inspired by residual neural networks, thus we term our models *residual deep Gaussian processes*. Instead of constructing a manifold-to-manifold GP layer directly, we represent it as a *Gaussian vector field* (GVF) combined with an exponential map. The former represents a displacement vector, a deviation from the identity map, or a *residual*, while the latter translates the input by the given displacement vector. The mean of a layer is always the output of the previous layer. We visualise this architecture in Figure 1, with sphere as the manifold of interest. Notably, by changing the last layer only, one can get manifold-valued or vector-valued deep GPs. As Section 3 will show, our residual deep Gaussian processes generalise the deep GP architecture of Salimbeni and Deisenroth (2017), perhaps the most successful architecture in the Euclidean case.

We examine residual deep GPs through synthetic and real-world experiments, demonstrating our models' superior performance over shallow geometry-aware GPs on tasks where complex data inherently lies on a manifold. Additionally, we show that our models offer prospective avenues for accelerating inference for inherently Euclidean data in the context of deep GPs. Our main focus is on hypersphere manifolds $\mathbb{S}_d$, due to their importance in key applications such as climate modelling and robotics, as well as their particularly simple structure that allows for more powerful and specialised GVFs (Robert-Nicoud et al., 2024). However, the applicability of our model extends to all Riemannian manifolds, including ones represented by meshes, indicating an even broader potential.

## 2 BACKGROUND

Mathematically, a Gaussian process (GP) is a real-valued random function $f$ whose marginals are jointly Gaussian. The same term is also used for the respective distribution over functions. For such $f$ there always exist a mean $\mu\colon X \to \mathbb{R}$ on the input domain $X$ of $f$ and a kernel $k\colon X \times X \to \mathbb{R}$ such that $f(\boldsymbol{x}) \sim \mathcal{N}(\mu(\boldsymbol{x}), k(\boldsymbol{x}, \boldsymbol{x}))$ for all $\boldsymbol{x} \subseteq X$. In this case, we write $f \sim \mathcal{GP}(\mu, k)$.

GPs define useful priors for learning functions from noisy observations $\boldsymbol{y} \in \mathbb{R}^n$ at given input locations $\boldsymbol{x} \subseteq X$ within the Bayesian framework. In fact, if the observation likelihood is assumed Gaussian $\boldsymbol{y} \mid f(\boldsymbol{x}) \sim \mathcal{N}(f(\boldsymbol{x}), \boldsymbol{\Sigma})$, then the posterior is a GP (Rasmussen and Williams, 2006) with

$$\mu_{f|\boldsymbol{y}}(\cdot) = \mu(\cdot) + k(\cdot, \boldsymbol{x})(k(\boldsymbol{x}, \boldsymbol{x}) + \boldsymbol{\Sigma})^{-1}(\boldsymbol{y} - \mu(\boldsymbol{x})), \tag{1}$$

$$k_{f|\boldsymbol{y}}(\cdot, \cdot') = k(\cdot, \cdot') - k(\cdot, \boldsymbol{x})(k(\boldsymbol{x}, \boldsymbol{x}) + \boldsymbol{\Sigma})^{-1}k(\boldsymbol{x}, \cdot'). \tag{2}$$

The input domain $X$ can, in principle, be any set. However, a good kernel $k\colon X \times X \to \mathbb{R}$ is necessary to define practical GP models on $X$. If $X = \mathbb{R}^d$, the most widely used kernels are from the Matérn family (Rasmussen and Williams, 2006), including, as a limiting case, the especially popular squared exponential kernel.[2] These kernels are attractive because they implement two natural inductive biases: (1) model behaviour should not change under any symmetries of the data—translations, in the case of $\mathbb{R}^d$—and (2) the unknown function possesses a certain degree of smoothness. It turns out that the Matérn family can be generalised to various non-Euclidean domains $X$ in such a way that it still implements (1) and (2). We now discuss such a generalisation to Riemannian manifolds[3].

---

[1] Nevertheless, some practical heuristics for this exist in the literature (Mallasto and Feragen, 2018).

[2] This kernel has many names; it is also known as the RBF, Gaussian, heat, or diffusion kernel.

[3] For more details, see Azangulov et al. (2024), Borovitskiy et al. (2020), and Hutchinson et al. (2021).

## 2.1 GAUSSIAN PROCESSES ON RIEMANNIAN MANIFOLDS

A principled way of generalising the family of Matérn kernels to Riemannian manifolds was proposed by Lindgren et al. (2011) based on the ideas dating back to Whittle (1963). Borovitskiy et al. (2020) showed that the resulting kernels can be represented as the following infinite series

$$k_{\nu,\kappa,\sigma^2}(x,x') = \frac{\sigma^2}{C_{\nu,\kappa}} \sum_{j=0}^{\infty} \Phi_{\nu,\kappa}(\lambda_j)\phi_j(x)\phi_j(x'), \quad \Phi_{\nu,\kappa}(\lambda) = \begin{cases} \left(\frac{2\nu}{\kappa^2} + \lambda\right)^{-\nu-\frac{d}{2}} & \nu < \infty \\ e^{-\frac{\kappa^2}{2}\lambda} & \nu = \infty \end{cases} \quad (3)$$

where $-\lambda_j, \phi_j$ are the eigenpairs of the Laplace–Beltrami operator on $X$, $d$ is the dimension of $X$, and $C_{\nu,\kappa}$ is a normalisation constant ensuring $\frac{1}{\text{vol}X} \int_X k_{\nu,\kappa,\sigma^2}(x,x) \, dx = \sigma^2$. The infinite sum must be truncated for computational tractability; nevertheless, the rapid decay of the coefficients $\Phi_{\nu,\kappa}(\lambda_j)$ makes this a sensible approximation with convergence guarantees (Rosa et al., 2023).

Such models were used, e.g., in robotics (Jaquier et al., 2022; 2024) and medical (Coveney et al., 2020) applications. Their vector field counterparts, to which we return later in the paper, were used for modelling wind velocities on the globe (Hutchinson et al., 2021; Robert-Nicoud et al., 2024).

These models tend to perform well, especially when data is scarce and uncertainty quantification is crucial, but they can struggle to capture complex irregular patterns. One potential way to improve on this is to consider *deep Gaussian processes*, which—in the Euclidean case—we now review.

## 2.2 DEEP GAUSSIAN PROCESSES AND APPROXIMATE INFERENCE

A deep Gaussian process $F$ is a composition $F = f^L \circ \cdots \circ f^1$ of multiple shallow GPs $f^l$ (Damianou and Lawrence, 2013). To allow for richer structure, the hidden *layers* $f^l$, for $1 \leq l \leq L - 1$, are typically *vector-valued* GPs $f^l \colon \mathbb{R}^d \to \mathbb{R}^d$, i.e. vectors of scalar-valued GPs stacked together and potentially correlated with each other (Álvarez et al., 2012). The resulting random function $F$ is itself not a GP. Thus, even for Gaussian likelihoods $p(\boldsymbol{y} \mid F(\boldsymbol{x}))$, inference for $F \mid \boldsymbol{y}$ is intractable.

To overcome this, various approximate inference techniques for deep GPs were proposed, perhaps the most popular being *doubly stochastic variational inference* (Salimbeni and Deisenroth, 2017). In it, the intractable posterior $F \mid \boldsymbol{y}$ is approximated in terms of the KL divergence metric $D_{\text{KL}}$ by the elements of a certain *variational family* of tractable distributions. This family itself consists of deep GPs whose layers are *sparse GPs* (Hensman et al., 2013; Titsias, 2009) which we now discuss.

Sparse GPs were originally proposed as a variational family for approximating shallow GPs. For them, approximate inference helps scale to big datasets or accommodate non-Gaussian likelihoods. Take some $f \sim \mathcal{GP}(\mu, k)$. A sparse GP $f_{\boldsymbol{z},\boldsymbol{m},\boldsymbol{S}}$ is a family of GPs parameterised by a set of $m$ *inducing locations* $\boldsymbol{z} \subseteq X$, as well as a mean vector $\boldsymbol{m} \in \mathbb{R}^m$ and a covariance matrix $\mathbf{S} \in \mathbb{R}^{m \times m}$ which determine the corresponding *inducing variable* distribution $q(\boldsymbol{u}) = \mathcal{N}(\boldsymbol{m}, \mathbf{S})$. Specifically,

$$p(f_{\boldsymbol{z},\boldsymbol{m},\boldsymbol{S}}(\cdot)) = \mathbb{E}_{\boldsymbol{u} \sim q(\boldsymbol{u})} \, p(f(\cdot) \mid \boldsymbol{u}, \boldsymbol{z}), \qquad q(\boldsymbol{u}) = \mathcal{N}(\boldsymbol{m}, \mathbf{S}), \qquad (4)$$

where $p(f(\cdot) \mid \boldsymbol{u}, \boldsymbol{z})$ is the prior $f$ conditioned on $f(\boldsymbol{z}) = \boldsymbol{u}$. Intuitively, $\boldsymbol{z}$ are the pseudo-inputs, $\boldsymbol{u}$ are random pseudo-observations, and $p(f_{\boldsymbol{z},\boldsymbol{m},\boldsymbol{S}}(\cdot))$ is a kind of pseudo-posterior. It is a GP with

$$\mu_{\boldsymbol{z},\boldsymbol{m},\boldsymbol{S}}(\cdot) = \mu(\cdot) + k(\cdot,\boldsymbol{z})k(\boldsymbol{z},\boldsymbol{z})^{-1}(\boldsymbol{m} - \mu(\boldsymbol{z})), \qquad (5)$$

$$k_{\boldsymbol{z},\boldsymbol{m},\boldsymbol{S}}(\cdot,\cdot') = k(\cdot,\cdot') - k(\cdot,\boldsymbol{z})k(\boldsymbol{z},\boldsymbol{z})^{-1}(k(\boldsymbol{z},\boldsymbol{z}) - \mathbf{S})k(\boldsymbol{z},\boldsymbol{z})^{-1}k(\boldsymbol{z},\cdot'). \qquad (6)$$

These can be readily generalised to the vector-valued setting, with $\boldsymbol{m} \in \mathbb{R}^{md}$ and $\mathbf{S} \in \mathbb{R}^{md \times md}$.

For a deep GP, the respective variational family is the distribution of a composition of sparse GPs:

$$F_{\boldsymbol{\theta}} = f_{\boldsymbol{z}^L,\boldsymbol{m}^L,\mathbf{S}^L} \circ \ldots \circ f_{\boldsymbol{z}^1,\boldsymbol{m}^1,\mathbf{S}^1}, \qquad \boldsymbol{\theta} = \left\{\boldsymbol{z}^l, \boldsymbol{m}^l, \mathbf{S}^l\right\}_{l=1}^{L}. \qquad (7)$$

The variational parameters $\boldsymbol{\theta}$ are found by minimising $D_{\text{KL}}(p(F \mid \boldsymbol{y}) \,\|\, p(F_{\boldsymbol{\theta}}))$, which is equivalent (Salimbeni and Deisenroth, 2017) to maximising the following evidence lower bound (ELBO)

$$\text{ELBO} = \sum_{i=1}^{n} \mathbb{E}_{F(x_i) \sim p(F_{\boldsymbol{\theta}}(x_i))} \log p(y_i | F(x_i)) - \sum_{l=1}^{L} D_{\text{KL}}(q(\boldsymbol{u}^l) \,\|\, p(\boldsymbol{u}^l)), \qquad (8)$$

where $q(\boldsymbol{u}^l) \sim \mathcal{N}(\boldsymbol{m}^l, \mathbf{S}^l)$ and $p(\boldsymbol{u}^l) \sim \mathcal{N}(\mu^l(\boldsymbol{z}^l), k^l(\boldsymbol{z}^l, \boldsymbol{z}^l))$. The second term can be computed exactly using the formula for $D_{\text{KL}}$ between two Gaussian vectors. The first term is intractable, but can be efficiently approximated by drawing a sample from $p(F_{\boldsymbol{\theta}})$—this can be done in a layerwise fashion, as Salimbeni and Deisenroth (2017) suggest—and subsampling the sum over a mini-batch of inputs $\boldsymbol{x}$. This way, optimisation proceeds by stochastic gradient descent.

## 3 RESIDUAL DEEP GAUSSIAN PROCESSES ON MANIFOLDS

In this section, we introduce the new model class of *residual deep Gaussian processes on manifolds*. It generalises the notion of deep GPs to Riemannian manifolds, allowing for the modelling of scalar- and vector-valued functions, vector fields, and functions taking values in the input manifold itself.

### 3.1 THE ARCHITECTURE

Let $X$ be a Riemannian manifold. The key challenge in building a deep Gaussian process $F$ on $X$ is finding a practical notion of manifold-to-manifold GPs $f^l$ to serve as its hidden layers:

$$F = f^L \circ f^{L-1} \circ \cdots \circ f^2 \circ f^1, \qquad f^l \colon X \to X \ \text{ for } \ 1 \le l \le L-1, \qquad (9)$$

where, to simplify exposition, we assume that the last layer $f^L$ is real-valued, although it can just as well be $X$-valued, vector-valued, or it can be a vector field, depending on the problem at hand.

While building a GP with inputs in $X$ amounts to finding an appropriate kernel $k \colon X \times X \to \mathbb{R}$, handling outputs in $X$ requires redefining the inherently Euclidean notion of a *Gaussian*. We aim to circumvent this difficulty. To explain how, we start by considering the popular Euclidean deep GP architecture of Salimbeni and Deisenroth (2017). There, each layer $f^l \colon \mathbb{R}^d \to \mathbb{R}^d$ is of the form

$$f^l(x) = x + g^l(x), \qquad (10)$$

where $g^l$ is a zero-mean GP. That is, each layer $f^l$ displaces its input $x$ by a *residual* vector $g^l(x) = f^l(x) - x$, the difference to the identity transform, much like a residual connection in neural networks (He et al., 2016), which is modelled by a GP. On a manifold $X \ne \mathbb{R}^d$, when $x \in X$ and $g^l(x) \in \mathbb{R}^d$, the addition operation in $f^l(x) = x + g^l(x)$ is undefined. However, there is a natural generalisation:

$$f^l(x) = \exp_x\big(g^l(x)\big). \qquad (11)$$

Here, $\exp_x \colon T_x X \to X$ is the *exponential map*, the canonical mapping of the *tangent space $T_x X$* at $x \in X$—i.e., the linear space of vectors tangent to $X$ at point $x$—back to $X$ itself. That is, a point $x$ is still displaced by the vector $g^l(x)$, but in a geometrically sound manner.

The beauty of Equation (11) is that it reduces modelling $f^l$ to modelling $g^l$. The latter is vector-valued, and thus compatible with the traditional notion of a Gaussian, making the problem conceptually much simpler. Still, a major technical difficulty remains: for different inputs $x$, the value $g^l(x)$ must lie in the different spaces $T_x X$. The mappings behaving like this are called *vector fields*, and their random Gaussian counterparts are called *Gaussian vector fields*, which we proceed to discuss.

### 3.2 KEY BUILDING BLOCKS: GAUSSIAN VECTOR FIELDS

A vector field on a manifold $X$ is a function that takes each $x \in X$ to an element of the tangent space $T_x X$ of $X$ at the point $x$. If $X$ is[4] a submanifold of $\mathbb{R}^D$—like the 2-sphere is a submanifold of $\mathbb{R}^3$—the difference between a vector field and a general vector-valued function on $X$ is very intuitive: in the latter, a vector attached to a point $x \in X$ can point in any direction, while in the former it must always be tangential to the manifold $X$ at $x$. This difference can be seen in Figure 2a, which features a vector-valued function on the left and an actual vector field on the right.

A Gaussian vector field (GVF) can be thus thought of as a vector-valued GP whose outputs always happen to be tangential vectors. This notion is rigorously formalised in the appendices of Hutchinson et al. (2021). However, for simplicity, we do not dwell on the formalism here. Instead, we proceed to discuss three practicable GVF constructions that have been put forward in recent research.

**Projected GVFs** Hutchinson et al. (2021) propose a simple idea, to build a GVF $g$ from any given vector-valued GP $\boldsymbol{h} \colon X \subset \mathbb{R}^D \to \mathbb{R}^D$ by *projecting* its outputs onto the appropriate tangent spaces. Such a projection $P_{(\cdot)} \colon \mathbb{R}^D \to T_{(\cdot)} X$ exists because, if $X$ is a submanifold of $\mathbb{R}^D$, then any tangent space $T_{(\cdot)} X$ can be identified with a linear subspace of $\mathbb{R}^D$. Thus, $g(x) = P_x \boldsymbol{h}(x)$ defines a random vector field (see Figure 2a), which turns out to be Gaussian because of the linearity of $P_{(\cdot)}$.

---

[4]*Nash embedding theorem* proves this is always the case for a large enough ambient space dimension $D \in \mathbb{N}$.

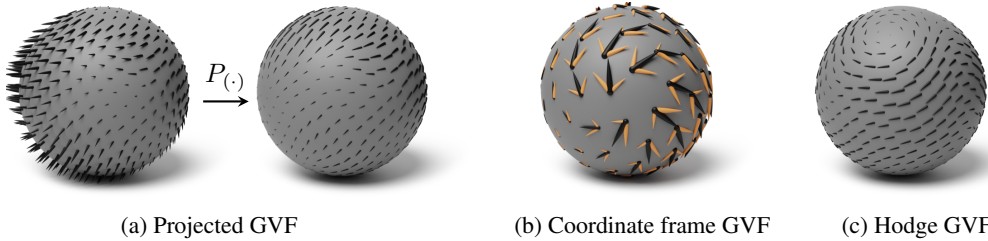

(a) Projected GVF        (b) Coordinate frame GVF      (c) Hodge GVF

Figure 2: Gaussian vector field constructions on the sphere. In (b), orange vectors depict the frame.

**Coordinate-frame-based GVFs**    Given any *coordinate frame* $\{e_i\}_{i=1}^d$—that is, a set of functions such that $\{e_i(\cdot)\}_{i=1}^d$ is a linear basis of $T_{(\cdot)}X$—and a vector-valued GP $\boldsymbol{h}\colon X \to \mathbb{R}^d$ with components $h_i\colon X \to \mathbb{R}$, the equation $g(x) = \sum_{i=1}^d h_i(x)e_i(x)$ defines a GVF. This is shown in Figure 2b.

**Hodge GVFs**    Most recently, Robert-Nicoud et al. (2024) extended the generalisation of Matérn GPs from Section 2.1 to the setting of vector fields on compact manifolds. They derive an analogue of Equation (3), representing the respective *Hodge Matérn kernels*, as an infinite series[5]

$$\boldsymbol{k}_{\nu,\kappa,\sigma^2}(x, x') = \frac{\sigma^2}{C_{\nu,\kappa}} \sum_{j=0}^{\infty} \Phi_{\nu,\kappa}(\lambda_j) s_j(x) \otimes s_j(x'). \tag{12}$$

Here, $s_j$ are the eigenfields of the Hodge Laplacian on $X$ that correspond to the eigenvalues $-\lambda_j$, $\otimes$ is the tensor product, and $\Phi_{\nu,\kappa}$ is exactly as in Equation (3). This family of kernels can be made more expressive by using different hyperparameters $\sigma^2$, $\kappa$, and $\nu$ for different *types* of eigenfields $s_j$: the *pure-divergence* $s_j$, the *pure-curl* $s_j$, and the harmonic $s_j$. The result is called *Hodge-compositional* Matérn kernels (Robert-Nicoud et al., 2024; Yang et al., 2023). For example, it can represent the inductive bias of divergence-free vector fields, i.e. vector fields having no "sinks" and "sources", like the wind velocity field at certain altitudes. This can be seen in Figure 2c.

The first two constructions are *universal*. This means that by choosing an appropriate $\boldsymbol{h}$, one can obtain any possible GVF. Although this might seem advantageous, this is also a major curse, as it is often unclear which particular $\boldsymbol{h}$ to take to get good inductive biases. What is more, simple solutions, such as $\boldsymbol{h}$ with IID components, may lead to undesirable artefacts (Robert-Nicoud et al., 2024). On the other hand, the third construction is *canonical*, in the same way as the Matérn family is canonical in the scalar Euclidean case, and it is based on the same simple and natural inductive biases.

Hodge GVFs seem to be the most attractive building blocks for deep GPs. However, although they are generally applicable *in theory*, Robert-Nicoud et al. (2024) only provide practical expressions for the eigenfields $s_n$ when $X$ is the circle $\mathbb{S}_1$, the 2-sphere $\mathbb{S}_2$, or any finite product of those. Thus, for the manifolds that go beyond this simple form, other GVF constructions have to be used. With this, we finish introducing our models, and proceed to discuss Bayesian inference for them.

## 3.3   Inference

Like their Euclidean counterparts, residual deep GPs constitute complex non-Gaussian priors, making exact Bayesian inference with them impossible. However, the doubly stochastic variational inference approach described in Section 2.2 is applicable to them after a few adjustments we detail below. What is more, for compact manifolds, the approach can be further modified to use certain interdomain inducing variables, which, as Section 4 shows, tends to offer superior performance.

**Doubly stochastic variational inference**    Consider the analogue of the variational family $p(F_{\boldsymbol{\theta}})$ in Equation (7) with the sparse GP layers $f_{\boldsymbol{z}^l,\boldsymbol{m}^l,\mathbf{S}^l}$ replaced by

$$f_{\boldsymbol{z}^l,\boldsymbol{m}^l,\mathbf{S}^l}(\cdot) = \exp_{(\cdot)}\big(g_{\boldsymbol{z}^l,\boldsymbol{m}^l,\mathbf{S}^l}(\cdot)\big), \tag{13}$$

where $g_{\boldsymbol{z}^l,\boldsymbol{m}^l,\mathbf{S}^l}$ are *sparse Gaussian vector fields*. Again, intuitively and in practice, treating the manifold $X$ as a submanifold of $\mathbb{R}^D$, GVFs can be thought of, and worked with, as a special kind of

---

[5]Robert-Nicoud et al. (2024) notice that sometimes, by analytically summing the terms corresponding to the same eigenvalue $\lambda_n$, the series convergence can be sped up by many orders of magnitude, improving efficiency.

vector-valued GPs. For $g_{\boldsymbol{z}^l, \boldsymbol{m}^l, \mathbf{S}^l}$, however, $\boldsymbol{z}^l = (z_1^l, \ldots, z_m^l)$ with inducing locations $z_j^l \in X$, and

$$\boldsymbol{m}^l = (\boldsymbol{m}_1^l, \ldots, \boldsymbol{m}_m^l), \qquad \boldsymbol{m}_j^l \in T_{z_j^l} X, \qquad \mathrm{im}(\mathbf{S}^l) \subseteq T_{z_1^l} X \times \ldots \times T_{z_m^l} X, \qquad (14)$$

where $\mathrm{im}$ denotes the *image* of a linear operator, and the last two constraints ensure that the random pseudo-observations $\boldsymbol{u} \sim \mathcal{N}(\boldsymbol{m}, \mathbf{S})$ are tangent vectors which lie in the appropriate tangent spaces.

To satisfy the aforementioned constraints during optimisation, one can represent

$$\boldsymbol{m}^l = P_{\boldsymbol{z}^l} \widetilde{\boldsymbol{m}}^l, \qquad \mathbf{S}^l = P_{\boldsymbol{z}^l} \widetilde{\mathbf{S}}^l P_{\boldsymbol{z}^l}^\top, \qquad P_{\boldsymbol{z}^l} = P_{z_1^l} \oplus \ldots \oplus P_{z_m^l}, \qquad (15)$$

with arbitrary $\widetilde{\boldsymbol{m}}^l \in \mathbb{R}^{mD}$, arbitrary positive semi-definite $\widetilde{\mathbf{S}}^l \in \mathbb{R}^{mD \times mD}$, and $P_z : \mathbb{R}^D \to T_z X$ denoting the projection of vectors in $\mathbb{R}^D$ onto the tangent space $T_z X$, as discussed in Section 3.2.

Instead of using such representations, one can fix a (locally) smooth frame and optimise the coefficients of $\boldsymbol{m}^l$ and $\mathbf{S}^l$ represented in this frame. In any case, one needs to make sure $z_j^l$ always remain on the manifold $X$ during optimisation, which can be done by using specialised libraries, such as PYMANOPT (Townsend et al., 2016) or GEOOPT (Kochurov et al., 2020). For low-dimensional manifolds, a fixed grid on $X$ or a set of cluster centroids can be an effective alternative to optimising $z_j^l$.

Finally, to approximate ELBO, we need to sample $F(x_i) \sim p(F_{\boldsymbol{\theta}}(x_i))$. As in Salimbeni and Deisenroth (2017), this can be done sequentially. Specifically, if $\hat{F}(x_i)$ denotes the desired sample, then

$$\hat{F}(x_i) = \hat{f}_i^L, \qquad \hat{f}_i^l = \exp_{\hat{f}_i^{l-1}}(g_{\boldsymbol{z}^l, \boldsymbol{m}^l, \mathbf{S}^l}(\hat{f}_i^{l-1})) \text{ for } 1 \leq l \leq L, \qquad \hat{f}_i^0 = x_i, \qquad (16)$$

and, given $\hat{f}_i^{l-1}$, each individual $g_{\boldsymbol{z}^l, \boldsymbol{m}^l, \mathbf{S}^l}(\hat{f}_i^{l-1})$ can be sampled in the usual manner.

**Interdomain inducing variables on manifolds**  On compact manifolds, an alternative variational family can be used that speeds up the inference and can often lead to better predictive performance. It is constructed by replacing the inducing locations $\boldsymbol{z}$ by *inducing linear functionals* $\boldsymbol{\zeta} = (\zeta_1, \ldots, \zeta_m)$. Each $\zeta_j$ takes in a vector field $g$ and outputs a real number. These $\boldsymbol{\zeta}$ define a sparse GVF through

$$p(g_{\boldsymbol{\zeta}, \boldsymbol{m}, \boldsymbol{S}}(\cdot)) = \mathbb{E}_{\boldsymbol{u} \sim q(\boldsymbol{u})} p(g(\cdot) \mid \boldsymbol{u}, \boldsymbol{\zeta}), \qquad q(\boldsymbol{u}) = \mathcal{N}(\boldsymbol{m}, \mathbf{S}), \qquad (17)$$

where $p(g(\cdot) \mid \boldsymbol{u}, \boldsymbol{\zeta})$ is the prior $g$ conditioned on $\boldsymbol{\zeta}(g) = \boldsymbol{u}$, where $\boldsymbol{\zeta}(g) = (\zeta_1(g), \ldots, \zeta_m(g))$. For example, linear functionals of the form $\zeta(g) = \langle g(z), e_i(z) \rangle_{T_z X}$—here, $\{e_i\}_{i=1}^d$ is a coordinate frame—can be used to recover the usual doubly stochastic variational inference considered above.

The mean and covariance of $g_{\boldsymbol{\zeta}, \boldsymbol{m}, \boldsymbol{S}}$ are given by Equation (5), with $\boldsymbol{z}$ replaced by $\boldsymbol{\zeta}$ and

$$k(\boldsymbol{\zeta}, \cdot) = \mathrm{Cov}(\boldsymbol{\zeta}(g), g(\cdot)), \qquad k(\boldsymbol{\zeta}, \boldsymbol{\zeta}') = \mathrm{Cov}(\boldsymbol{\zeta}(g), \boldsymbol{\zeta}(g)), \qquad (18)$$

see, for example, Lázaro-Gredilla and Figueiras-Vidal (2009) and van der Wilk et al. (2020).

Now, if the kernel of $g$ can be expressed as $\sum a_j \phi_j(x) \otimes \phi_j(x')$ where $\{\phi_j\}$ is an orthonormal basis—this is obviously so for Hodge GVFs, but is also often the case for other GVFs on compact manifolds—the inducing functionals $\zeta_j(\cdot) = \langle \cdot, \phi_j \rangle_{L^2} / a_j$ yield very simple covariance matrices

$$k(\zeta_j, \cdot) = \phi_j(\cdot), \qquad k(\zeta_i, \zeta_j) = \delta_{i,j}/a_i. \qquad (19)$$

In particular, $k(\boldsymbol{\zeta}, \boldsymbol{\zeta})$ is diagonal, making it trivial to invert. Dutordoir et al. (2020) report that this can yield significant acceleration in practice. For residual deep GPs, this phenomenon affects every individual layer, thus making the cumulative effect even more pronounced. We refer the reader to Appendix B for further practical and theoretical considerations regarding this variational family.

**Posterior mean, variance, and samples**  Expectations $\mathbb{E}\, F_{\boldsymbol{\theta}}(x)$ and variances $\mathrm{Var}\, F_{\boldsymbol{\theta}}(x)$ of the approximate posterior $F_{\boldsymbol{\theta}}$ cannot be computed exactly. Instead, they can be estimated by appropriate Monte Carlo averages, with Equation (16) providing a way to sample $F_{\boldsymbol{\theta}}(x)$. However, since these estimates ignore the correlation between $F_{\boldsymbol{\theta}}(x)$ and $F_{\boldsymbol{\theta}}(x')$, they are not continuous as functions of $x$. When continuity or differentiability of $\mathbb{E}\, F_{\boldsymbol{\theta}}(x)$ and $\mathrm{Var}\, F_{\boldsymbol{\theta}}(x)$ are desirable, another method can be used. The key idea in this case is to draw (approximate) samples from $F_{\boldsymbol{\theta}}(\cdot)$ which happen to be actual functions, for example linear combinations of some analytic basis functions or compositions of such. This can be done by applying the *pathwise conditioning* of Wilson et al. (2020) and Wilson et al. (2021) in a sequential manner, akin to Equation (16). This approach is useful for visualisation, performance metric estimation, and for working with downstream quantities, such as acquisition functions in Bayesian optimisation, for which differentiability is key for efficiently finding their maxima.

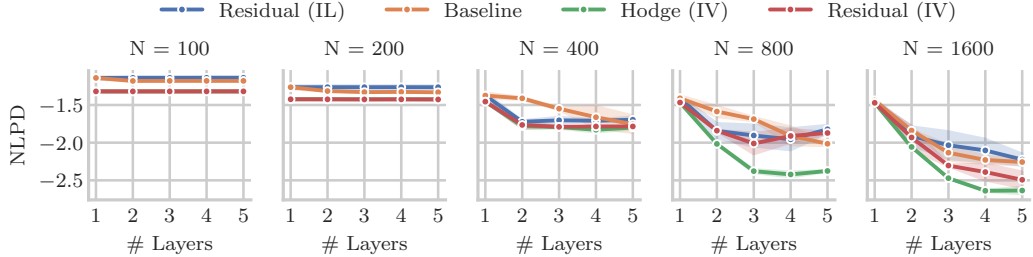

Figure 3: NLPD of different residual deep GP variants and the baseline model, on the regression problem for the synthetic benchmark function visualised in Figure 4a. Different subplots correspond to different training set sizes $N$. The solid lines represent the mean, while the shaded areas represent the $\pm 1$ standard deviation region around it. All statistics are computed over 5 randomised runs.

## 4 EXPERIMENTS

We begin this section by examining how various GVF and variational family choices impact the regression performance of residual deep GPs in synthetic experiments, as discussed in Section 4.1. Throughout, we compare our models to a baseline with Euclidean hidden layers. Next, in the robotics-inspired experiments of Section 4.2, we demonstrate that residual deep GPs can significantly enhance Bayesian optimisation on a manifold when the optimised function is irregular. Following this, in Section 4.3, we show state-of-the-art predictive and uncertainty calibration performance of residual deep GPs in wind velocity modelling on the globe, achieving interpretable patterns even at low altitudes where data is more complex and irregular. Finally, in Section 4.4, we explore potential avenues for using residual deep GPs to accelerate inference on inherently Euclidean data.

### 4.1 SYNTHETIC EXAMPLES

**Setup**  Deep GPs bear the promise of outperforming their shallow counterparts in modelling complex, irregular functions. To test this, we construct a benchmark function $f^*$ on the 2-sphere $\mathbb{S}_2$ with multiple singularities, which is visualised in Figure 4a. We take $N \in \{100, 200, 400, 800, 1600\}$ training inputs $\boldsymbol{x}$ on a Fibonacci lattice on $\mathbb{S}_2$ and put $\boldsymbol{y} = f^*(\boldsymbol{x}) + \boldsymbol{\varepsilon}$, $\boldsymbol{\varepsilon} \sim \mathcal{N}(\boldsymbol{0}, 10^{-4}\mathbf{I})$. Then, we regress $f^*$ from $\boldsymbol{x}$ and $\boldsymbol{y}$. On this problem, we compare different modifications of the residual deep GPs amongst themselves and to a baseline, in terms of the negative log predictive density (NLPD) and the mean squared error (MSE) metrics on the test set of 5000 points, also on a Fibonacci lattice. All runs are conducted 5 times with different random seeds for the observation noise $\boldsymbol{\varepsilon}$.

**Models**  The baseline is a deep GP with Euclidean (rather than manifold-to-manifold) layers. It is constructed by composing a vector-valued Matérn GP whose signature is $X \to \mathbb{R}^3$ with the Euclidean deep GP of Salimbeni and Deisenroth (2017) on $\mathbb{R}^3$. For the residual deep GPs, we consider two different types of GVFs, projected and Hodge, and two types of variational families, the one based on inducing locations (IL) and the one based on interdomain variables (IV). To ensure comparability, we match the number of optimised parameters between models as closely as possible.

**Results**  The NLPD values are presented in Figure 3. The MSE values exhibit the same trends and can be found in Figure 8 in Appendix A. We observe three key patterns. First, residual deep GPs are never worse than their shallow counterparts, recovering the single-layer solution when data is sparse. Second, as data becomes more abundant and thus captures more complexity of $f^*$, residual deep GPs outperform the shallow GPs. Third, the IV variational family almost always improves over the IL one, and the best model—with considerable margin—is obtained by combining Hodge GVFs with the IV variational family. The residual deep GP based on projected GVFs and using the IV variational family is the second best model, which still outperforms the baseline in most cases.

### 4.2 GEOMETRY-AWARE BAYESIAN OPTIMISATION

**Motivation**  GPs are a widely used model class for Bayesian optimisation, a technique for optimising expensive-to-evaluate black-box functions that leverages uncertainty estimates to balance exploration and exploitation (Shahriari et al., 2016). In robotics, such problems arise, for example, when a control policy needs to be fine-tuned to a specific real-world environment. This task

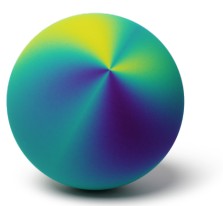 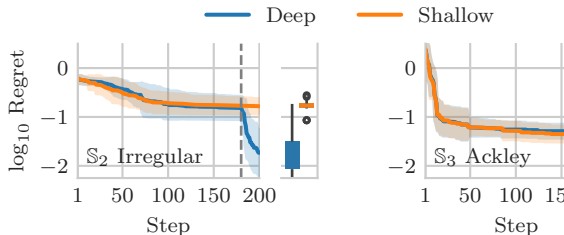

(a) Irregular benchmark function.     (b) Bayesian optimisation performance.

Figure 4: The irregular benchmark function, and Bayesian optimisation performance comparison. The target functions for Bayesian optimisation are: the aforementioned benchmark function, modified to have a single global minimum ($\mathbb{S}_2$ Irregular), and the smooth Ackley function on the 3-sphere ($\mathbb{S}_3$ Ackley). In (b), the solid lines represent the median regret, while the shaded areas around them span $\pm 1$ standard deviation. The statistics are computed over 15 randomised runs.

was shown to benefit from treating the optimisation space as a manifold and using geometry-aware Gaussian processes to drive Bayesian optimisation (Jaquier et al., 2022). The functions upon which the technique was tested are rather regular, which is not always the case in reality, especially when dealing with increasingly complex systems. Motivated by this challenge, we explore if residual deep GPs can offer improved performance in optimising complex irregular functions on manifolds.

**Setup** We consider two target functions to optimise. The first is the irregular function from Section 4.1, visualised in Figure 4a, modified to have only one global minimum. The second is the much more regular Ackley function, projected onto $\mathbb{S}_3$, one of the benchmarks in Jaquier et al. (2022). In each Bayesian optimisation run, we perform the first 180 iterations using a shallow geometry-aware GP, followed by 20 iterations using a residual deep GP—both employing the *expected improvement* acquisition function (see, e.g., Frazier (2018)). In this experiment, we showcase the coordinate-frame-based GVFs, as described in Appendix A. We do not use deep GPs in the initial iterations because, as intuition suggests and Section 4.1 affirms, deep GPs start outperforming their shallow counterparts only when data becomes more abundant. Although deep GPs show comparable performance even for small datasets, training them is more computationally demanding, making their use less efficient in the early stages of optimisation. We repeat each run 15 times to account for the stochasticity of initialisation, optimisation of the acquisition function, and training of GP models.

**Results** The optimisation performance, measured in terms of the logarithm of regret, is reported in Figure 4b. We find that residual deep GPs significantly improve performance in the Bayesian optimisation of the irregular function. Specifically, switching to a residual deep GP in the latter stages of optimisation greatly, and often immediately, reduces the gap between the true optimum and the found optimum. This trend is consistent across most runs, with only one outlier showing no improvement due to insufficient data collection near the singularity during the shallow GP phase. In contrast, for the Ackley function, we observe no substantial difference in performance between the two methods: both approaches replicate results from Jaquier et al. (2022), with nearly identical median regret trajectories. This outcome aligns with our expectations, since the region around the minimum, explored during the initial 180 iterations, is smooth and thus modelled equally well by both deep and shallow models.

## 4.3 WIND INTERPOLATION ON THE GLOBE

**Motivation** Non-Euclidean geometry has a particularly pronounced effect on vector fields, such as wind velocity fields on the globe, more so than on scalar functions. For instance, the famous *hairy ball theorem* states that a smooth vector field *must* always have a zero somewhere on the 2-sphere, i.e. there must always be a location where wind does not blow. Wind interpolation is thus an attractive use case for geometry-aware Gaussian vector fields, where they have been shown to perform well (Hutchinson et al., 2021; Robert-Nicoud et al., 2024). Here, we show that residual deep GPs can improve the performance of probabilistic wind velocity models when the data contains complex irregular patterns, which naturally occur in wind fields at lower altitudes.

**Setup** We consider the task of interpolating the monthly average wind velocity from the ERA5 dataset (Hersbach et al., 2023), from a set of locations on the Aeolus satellite track (Reitebuch,

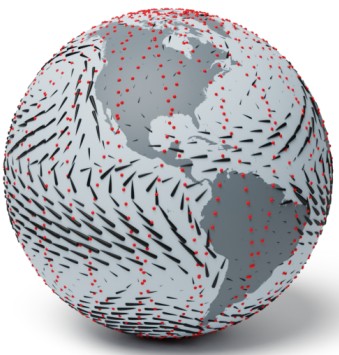
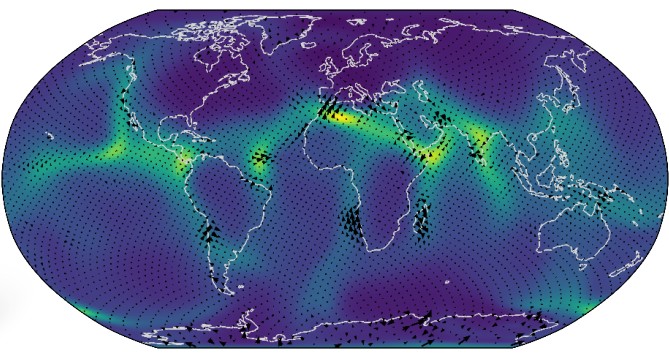

(a) The ground truth wind velocities as black arrows and the training locations along the Aeolus satellite track as red points.

(b) Difference between the prediction and the ground truth wind velocities, shown as black arrows, and the predictive uncertainty, shown using a colour scale from purple (lowest) to yellow (highest), for a 3-layer residual deep GP and wind velocities for July 2010, at 0.1 km altitude.

Figure 5: Using residual deep GPs for probabilistic wind velocity modelling on the surface of Earth.

2012), simulating a practical setting of a weather-analysing satellite. We use ERA5 data from January to December 2010, as in Robert-Nicoud et al. (2024), and sample the Aeolus track locations every minute for a 24-hour period from 9:00 am, January 1st, 2019. We choose a 24-hour period instead of a 1-hour period, as in Hutchinson et al. (2021), because in that time frame, the satellite produces a denser set of observations, crucial for capturing the complexity of wind behaviour at low altitudes. The ground truth vector field and the input locations are visualised in Figure 5a. To assess how decreasing regularity of data associated with decreasing altitude affects our model, we consider data at three altitudes: approximately 5.5 km, 2 km, and 0.1 km. In our models, we use Hodge GVFs in hidden layers and as the last layer, and interdomain inducing variables for inference.

**Results** We report regression performance, in terms of NLPD, in Figure 6; MSE follows similar trends and can be found in Figure 11 in Appendix A. We find that residual deep GPs improve upon the state-of-the-art shallow Hodge GVFs (1-layer models in the plots, Robert-Nicoud et al. (2024))

both in prediction quality and uncertainty calibration, as evidenced by the significantly lower NLPD and MSE. Furthermore, Figure 5b shows that the uncertainty estimates of our deep model at the lowest altitude are interpretable. Indeed, regions of high uncertainty follow regions of irregular wind currents, such as boundaries where multiple currents meet or continental boundaries, as well as areas of seasonally high winds, such as India during the peak monsoon season. At the same time, low uncertainty is assigned to regions with constant-like currents. This is unachievable for shallow GVFs since their posterior covariance depends only on the *locations* of the observations, which are rather uniformly dense in our setup, and not on the observations themselves. Additionally, Figure 5b shows predictive uncertainty to be well-calibrated, with areas of highest error corresponding to regions of high uncertainty.

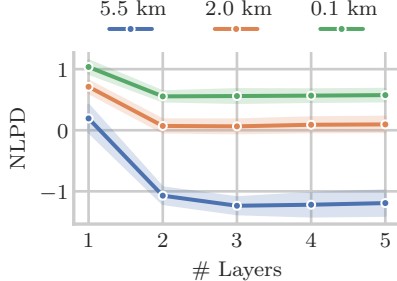

Figure 6: NLPD of residual deep GPs on the wind modelling task across three altitude levels. Solid lines give the mean NLPD, while the shaded regions around it span $\pm 1$ standard deviation. Statistics are computed from 12 runs—one for each month of 2010.

### 4.4 ACCELERATING INFERENCE FOR EUCLIDEAN DATA

**Motivation** Inspired by their connection to infinitely wide neural networks, Dutordoir et al. (2020) showed that geometry-aware GPs on hyperspheres can be applied to inherently Euclidean data to accelerate variational inference. Specifically, they reported that approximate variational inference

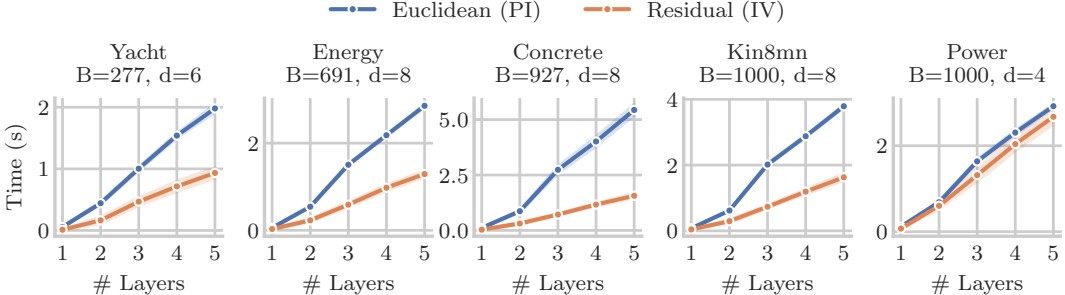

Figure 7: Wall clock time taken by one training step of Euclidean deep GPs with inducing locations, and residual deep GPs with interdomain variables. We consider 5 UCI datasets, with dimension $d$ and batch size $B$. Solid lines show the mean, computed by averaging over 100 training steps, while the shaded areas span $\pm 1$ standard deviation. However, they are often too narrow to be visible.

using the shallow analogue of the interdomain inducing variables applied to data mapped[6] from $\mathbb{R}^d$ to the proxy manifold of $\mathbb{S}_d$ can be significantly faster than inducing-location-based variational inference for a Euclidean GP on the original data, while achieving competitive predictive performance. We investigate whether this result can be extended to the case of deep Gaussian processes.

**Setup** As Euclidean data, we use the same UCI datasets as Dutordoir et al. (2020), and we use the same mapping from $\mathbb{R}^d$ to the proxy manifold $\mathbb{S}_d$. We use projected GVFs to accommodate arbitrarily-dimensional hyperspheres. Overall, our experimental setup follows Dutordoir et al. (2020), except that working with deep models required us to optimise ELBO directly instead of marginalising out the variational mean $m$ and covariance $\mathbf{S}$ as in Titsias (2009). Additionally, because of high memory requirements of L-BFGS (Nocedal, 1980) arising from the lack of marginalisation and depth, we switch to Adam (Kingma and Ba, 2015). We use a single Intel i7-13700H CPU.

**Results** We compare the variational inference speed, measured by wall-clock time for a single optimisation step, in Figure 7. The advantage for shallow 1-layer GPs increases significantly with more layers, offering a considerable edge in deep models. However, predictive performance comparisons in Figure 17 in Appendix A do not show such an optimistic picture: Euclidean deep GPs always outperform residual deep GPs with the same number of layers in terms of NLPD and MSE. Possibly, this might be due to the aforementioned differences in optimisation. Also, our choice of the mapping from $\mathbb{R}^d$ to a proxy manifold and the choice of the proxy manifold itself might be overly simplistic and thus hinder performance. We hypothesise that better mappings or optimisation could potentially make the tested approach a more efficient alternative to Euclidean deep GPs. Achieving this, however, will require further work, which is beyond the scope of this paper.

## 5 CONCLUSION

In this paper, we proposed a novel model class of *residual deep Gaussian processes* on manifolds. We reviewed practical Gaussian vector field constructions for building their hidden layers and discussed two variational inference techniques, including one tailored to the structure of Gaussian vector fields on compact manifolds and based on interdomain inducing variables. We evaluated our models in synthetic experiments, examining the impact of Gaussian vector field and variational family choices. These experiments supported favouring Hodge Gaussian vector fields and interdomain inducing variables. They also demonstrated that increasing the number of layers virtually never degrades our models' performance, though it can quickly saturate and plateau. We hypothesize that larger datasets will slow saturation, necessitating more layers' complexity. However, we leave this for future work to explore. In a robotics-motivated stylised experiment, our models significantly enhanced Bayesian optimisation for an irregular function on the sphere. For probabilistic interpolation of wind velocities, we achieved state-of-the-art performance, surpassing the recently proposed shallow Hodge Gaussian vector fields. Finally, we showed interdomain inducing variables to be superior in terms of inference time, compared to doubly stochastic variational inference for Euclidean deep Gaussian processes. This indicates potential future benefits for Euclidean data if suitable mappings from manifold data to proxy manifolds are found. We believe residual deep Gaussian processes will provide a powerful toolset for applications in climate modelling, robotics, and beyond.

---

[6]Mapping $\mathbb{R}^d \ni \boldsymbol{x} \mapsto (x_1, .., x_d, b)/\|(x_1, .., x_d, b)\| \in \mathbb{S}_d$, where $b \in \mathbb{R}$ is a *bias* term, $\boldsymbol{x} = (x_1, .., x_d)$.

ACKNOWLEDGMENTS

VB was supported by ELSA (European Lighthouse on Secure and Safe AI) funded by the European Union under grant agreement No. 101070617. KW thanks Edoardo Ponti for his mentorship. The Blender rendering scripts we used for plotting were adapted from Terenin (2022).

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

## A  ADDITIONAL EXPERIMENTAL DETAILS

### A.1  IMPLEMENTATION

**Efficient kernel evaluation with the addition theorem**  In our implementation of manifold Matérn kernels, we utilise the *addition theorem for spherical harmonics* (De Vito et al., 2021; Dutordoir et al., 2020) to accelerate kernel computation.[7] On $\mathbb{S}_d$, the eigenfunctions of the Laplace–Beltrami operator are known to be certain special functions called *spherical harmonics*. The addition theorem gives a relation between all spherical harmonics corresponding to the $(k+1)$-th largest eigenvalue of the negative Laplace–Beltrami operator $-\Delta$, denoted $\{\phi_{k,j}\}_{j=1}^J$, and Gegenbauer polynomials $C_k^{(\alpha)}$, other special functions:

$$\sum_{j=1}^{J} \phi_{k,j}(x)\phi_{k,j}(x') = c_{k,d}C_k^{(\alpha)}(x \cdot x') \tag{20}$$

with the dot product computed after embedding $\mathbb{S}_d$ in $\mathbb{R}^{d+1}$ as the unit sphere centred at the origin, $\alpha = \frac{d-1}{2}$, and $c_{k,d}$ some known absolute constants. Thus, when computing the scalar Matérn kernel on $\mathbb{S}_d$, we truncate the infinite sum in Equation (3) to include all spherical harmonics up to the $(K+1)$-th eigenvalue, and apply Equation (20). This gives the following formula

$$k_{\nu,\kappa,\sigma^2}(x,x') = \frac{\sigma^2}{C_{\nu,\kappa}} \sum_{k=0}^{K} \Phi_{\nu,\kappa}(\lambda_k)c_{k,d}C_k^{(\alpha)}(x \cdot x'). \tag{21}$$

In case of the Hodge Matérn kernel on $\mathbb{S}_2$, we also apply addition theorem, only noting that here $\phi_{k,j}$ are replaced with the normalised vector spherical harmonics: $\frac{\nabla \phi_{k,j}}{\sqrt{\lambda_k}}$ for the divergence-only kernel, and $\frac{\star \nabla \phi_{k,j}}{\sqrt{\lambda_k}}$ for the curl-only kernel, where $\star$ is the Hodge star operator (Robert-Nicoud et al., 2024). In this case, $c_{k,d} = \frac{k+\alpha}{\alpha}$. Thus,

$$k_{\nu,\kappa,\sigma^2}^{\mathrm{div}}(x,x') = \frac{\sigma^2}{C_{\nu,\kappa}^{\mathrm{div}}} \sum_{k=0}^{K} \frac{\Phi_{\nu,\kappa}(\lambda_k)}{\lambda_k}\frac{k+\alpha}{\alpha}(\nabla_x \otimes \nabla_{x'})C_k^{(\alpha)}(x \cdot x') \tag{22}$$

$$k_{\nu,\kappa,\sigma^2}^{\mathrm{curl}}(x,x') = \frac{\sigma^2}{C_{\nu,\kappa}^{\mathrm{curl}}} \sum_{k=0}^{K} \frac{\Phi_{\nu,\kappa}(\lambda_k)}{\lambda_k}\frac{k+\alpha}{\alpha}(\star\nabla_x \otimes \star\nabla_{x'})C_k^{(\alpha)}(x \cdot x'). \tag{23}$$

The Hodge-compositional kernel, which we typically use, is the sum $k_{\nu,\kappa_1,\sigma_1^2}^{\mathrm{div}} + k_{\nu,\kappa_2,\sigma_2^2}^{\mathrm{curl}}$.

**Accelerated training with whitened inducing variables**  In all the models we test, approximate inference requires a variational mean vector $m$ and a variational covariance matrix $\mathbf{S}$ that parameterise $q(u) = \mathcal{N}(m, \mathbf{S})$.[8] However, to accelerate convergence during training, instead of working with the inducing variables $u$ directly, we work with *whitened* inducing variables $u' = \mathbf{L}^{-1}u$, where $\mathbf{L}$ is the lower Cholesky factor of $k(z, z)$ or $k(\zeta, \zeta)$ (Matthews, 2016). Thus, in practice, denoting the whitened variational mean and covariance of $q(u')$ as $m'$ and $\mathbf{S}'$, we use a modified version of Equation (5):

$$\mu_{z,m',\mathbf{S}'}(\cdot) = \mu(\cdot) + k(\cdot, z)k(z, z)^{-1}(\mathbf{L}m' - \mu(z)) \tag{24}$$

$$= \mu(\cdot) + k(\cdot, z)\mathbf{L}^{-\top}(m' - \mathbf{L}^{-1}\mu(z)), \tag{25}$$

$$k_{z,m',\mathbf{S}'}(\cdot, \cdot') = k(\cdot, \cdot') - k(\cdot, z)k(z, z)^{-1}(k(z, z) - \mathbf{L}\mathbf{S}'\mathbf{L}^{\top})k(z, z)^{-1}k(z, \cdot') \tag{26}$$

$$= k(\cdot, \cdot') - k(\cdot, z)\mathbf{L}^{-\top}(\mathbf{I} - \mathbf{S}')\mathbf{L}^{-1}k(z, \cdot'). \tag{27}$$

### A.2  MODELS

**Mean and kernel**  In all models, we equip constituent GPs with a zero mean and an appropriate variant of the Matérn kernel, initialised with length scale $\kappa = 1$. For kernels of output layers we

---

[7]The analogues of this theorem hold for many other manifolds, see, e.g., Azangulov et al. (2024).

[8]$\mathbf{S}$ is parametrised by its lower Cholesky factor to ensure positive definiteness during optimisation.

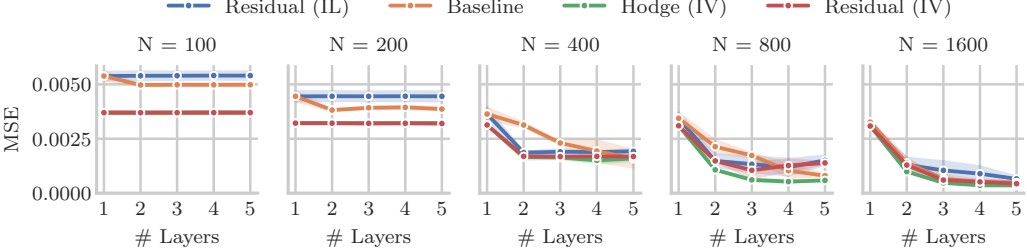

Figure 8: MSE of different residual deep GP variants and the baseline model, on the regression problem for the synthetic benchmark function in Figure 4a. Different subplots correspond to different training set sizes $N$. The solid lines represent the mean, while the shaded areas represent the $\pm\sigma$ region around it, where $\sigma$ is the standard deviation, and all statistics were computed over 5 randomised runs.

initialise the variance to $\sigma^2 = 1.0$, while for kernels in hidden layers of an $L$-layer deep GPs we set $\sigma^2 = \frac{10^{-4}}{L-1}$ at the start of training. In Section 4.1, Section 4.4, and Section 4.3 we initialise the smoothness parameter to $\nu = \frac{3}{2}$, while in Section 4.2 we set it to $\nu = \frac{5}{2}$ to replicate the setup in Jaquier et al. (2022). We optimise the smoothness of the manifold Matérn kernels during training, owing to their differentiability with respect to $\nu$, except in Section 4.2, where we fix $\nu$ to match the setup in Jaquier et al. (2022). Wherever we employ Hodge GVFs we use the Hodge compositional kernel, with separate $\nu, \kappa, \sigma^2$ for the curl-free and divergence-free parts. In models utilising interdomain variables, we use the same number of spherical harmonics for the kernel and inducing variables, as per our discussion in Appendix B.

**Vector-valued GPs**   We model vector-valued GPs as a set of *independent* scalar-valued GPs stacked into a vector. We utilise this construction in Euclidean deep GPs and in residual deep GPs with projected GVFs.

**Inducing locations**   Following Salimbeni and Deisenroth (2017), for all models utilising the variational family based on inducing locations $z^l$, we initialise $z^l$ for every layer to be the centers of the clusters found via k-means clustering of training data. In residual deep GPs, we further project these locations onto the sphere, and we do not optimise them during training. In Euclidean deep GPs, we do not normalise the inducing locations and optimise them jointly with all other parameters.

**Approximation in training and evaluation of deep models**   In all experiments, to approximate the ELBO in deep models during training, we use 3 samples from the posterior. In evaluation, we use 10 samples from the posterior to approximate the MSE and NLPD. In visualisation, we also use ten samples from the posterior to approximate the predictive mean and standard deviation.

## A.3   SYNTHETIC EXPERIMENTS

**Data**   To examine the influence of data density on model performance in a controlled manner, we generate the training sets as approximately uniform grid of points on the 2-sphere $\mathbb{S}_2$. This is done using the Fibonacci lattice method, which, for a grid of $n$ points, gives the colatitude and longitude of the $i$-th point as

$$\text{colatitude} = \arccos\left(1 - \frac{2i+1}{n}\right), \qquad \text{longitude} = \frac{2\pi i}{\phi}, \qquad \phi = \frac{1+\sqrt{5}}{2}. \qquad (28)$$

Because this method gives consistent coverage of $\mathbb{S}_2$ we also use it to generate the test set of 5000 points. We define the benchmark function $f^*: \mathbb{S}_2 \to \mathbb{R}$ by

$$f^*(x) = (Y_{2,3} \circ \varphi)(x) + (Y_{1,2} \circ \varphi \circ R)(x), \qquad (29)$$

where

$$Y_{2,3}(\theta, \phi) = \sqrt{\frac{105}{32\pi}} \sin^3\theta \sin(3\phi), \tag{30}$$

$$Y_{1,2}(\theta, \phi) = \sqrt{\frac{15}{8\pi}} \sin\theta \sin(2\phi) \tag{31}$$

$$R(x) = (x_1, -x_3, x_2) \tag{32}$$

$$\varphi(x) = (\text{atan2}(x_2, x_1), \arccos(x_3)), \tag{33}$$

with $x$ being an element of $\mathbb{S}_2$ embedded into $\mathbb{R}^3$ as the unit sphere centred at the origin. $Y_{2,3}, Y_{1,2}$ are spherical harmonics, smooth functions of their parameters. Singularities in $f^*$ are caused by the composition with $\varphi$, which converts $x$ from Cartesian to spherical coordinates, but swaps the positions of the colatitude $\theta$ and longitude $\phi$. The function $Y_{2,3} \circ \varphi$ has singularities at the poles, while the function $Y_{1,2} \circ \varphi \circ R$ has singularities around the equator.

**Variational parameters** To make the comparison between models as fair as possible, we set the number of inducing variables for each model in such a way that all models have almost the same total number of optimisable parameters. Specifically, we use the following formulae for the number of variational parameters $\alpha$ in a hidden layer, given that each GVF has $m$ inducing variables

$$\alpha_{\text{hodge}} = \underbrace{(m^2 + m)/2}_{\text{covariance}} + \underbrace{m}_{\text{mean}} \tag{34}$$

$$\alpha_{\text{euclidean}} = \alpha_{\text{projected}} = 3 \cdot ((m^2 + m)/2 + m) \tag{35}$$

In projected GVFs of the residual deep GPs and in vector-valued GPs of the baseline models, we equip each scalar GP with 49 inducing variables, which corresponds to spherical harmonics up to the 7-th negative eigenvalue. For Hodge GVFs, we use 70 interdomain inducing variables, corresponding to vector spherical harmonics up to the 6-th negative eigenvalue of the Hodge Laplacian. Despite this, residual deep GPs with Hodge GVFs are still at a disadvantage: a single Hodge GVF has 2555 variational parameters and $2 \cdot 3$ kernel parameters, a single projected GVF and manifold vector-valued GP has 3822 variational parameters and $3 \cdot 3$ kernel parameters, while one Euclidean vector-valued GP has 3822 variational parameters and $3 \cdot 4$ kernel parameters (we use automatic relevance determination (ARD), we get 4 as the sum of 3 length scale parameters and 1 prior variance parameter). Furthermore, we optimise the inducing locations in the Euclidean vector-valued GPs of the baseline model, so that each of them have an additional $3 \cdot 49$ optimisable parameters.

**Training and evaluation** We optimise all models using the Adam optimiser (Kingma and Ba, 2015) for 1000 iterations with learning rate set to 0.01.

**Additional results** The MSE comparison is presented in Figure 8. Figure 9 and Figure 10 show an extended comparison between the baseline model and Hodge residual deep GP with up to 10 layers and 10 randomised runs.

## A.4 GEOMETRY-AWARE BAYESIAN OPTIMISATION

**Data** To obtain an irregular function $f^*$ on $\mathbb{S}_2$, we modify the target function from Section 4.1 to have only one global minimum near a singularity point. Specifically, $f^*$ was defined by

$$f^*(x) = (Y_{2,3} \circ \varphi)(x) \cdot (x_3 + 1) \cdot (1 - \arccos(x_3)), \tag{36}$$

where $Y_{2,3}$ and $\varphi$ are as in Appendix A.3. The absence of $Y_{1,2}$ removes the singularities around the equator, while the added scaling factors create a minimum at the north pole and ensure it is global.

**Models** After a preliminary examination, we found that differences between GVFs variants on this task were not significant. Nevertheless, in Figure 4b we present the models that performed best in this examination: in the left subplot, a 2-layer residual deep GP using coordinate-frame GVFs and inducing locations; in the right subplot, a 3-layer residual deep GP using projected GVFs and inducing locations. We set the hyper-priors for the shallow model according to Jaquier et al. (2022). We do not use hyper-priors for the deep models in this experiment.

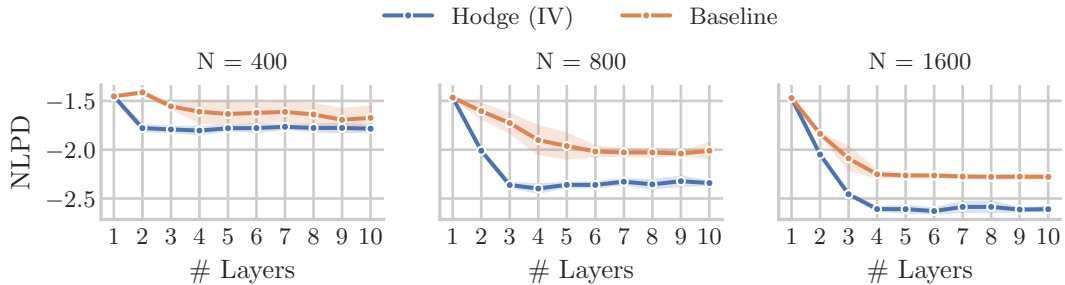

Figure 9: NLPD of the baseline model and Hodge residual deep GP on the regression problem for the synthetic benchmark function in Figure 4a. Different subplots correspond to different training set sizes $N$. The solid lines represent the mean, while the shaded areas represent the $\pm\sigma$ region around it, where $\sigma$ is the standard deviation; all statistics are computed over 10 randomised runs.

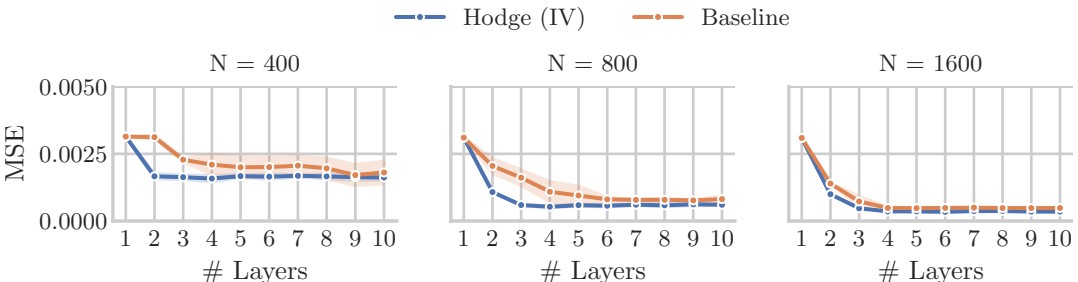

Figure 10: MSE of the baseline model and Hodge residual deep GP on the regression problem for the synthetic benchmark function in Figure 4a. Different subplots correspond to different training set sizes $N$. The solid lines represent the mean, while the shaded areas represent the $\pm\sigma$ region around it, where $\sigma$ is the standard deviation; all statistics are computed over 10 randomised runs.

**Optimisation process**  Replicating the setup of Jaquier et al. (2022), we begin each Bayesian optimisation by sampling 5 initial observations uniformly at random on the hypersphere. At each step, we minimise the expected improvement acquisition function using the first-order geometry-aware gradient optimisation implemented in PYMANOPT (Townsend et al., 2016). We approximate the expected improvement acquisition function for deep models using Monte Carlo averages driven by pathwise sampling, as described at the very end of Section 3.3. After each optimisation step, we reinitialise the model and fit it to the data for 500 iterations, using the Adam optimiser (Kingma and Ba, 2015) with a learning rate of 0.01 for the deep model, and the BFGS optimiser for the shallow model.

**Additional results and analysis**  In Figure 4b, we see that the variance of the logarithm of regret increases after switching from the shallow GP to the residual deep GP. This increase can be explained by variation in the initial 180 points acquired by the shallow GP between independent runs. The locations of these points determine the quality of fit of the deep GP. Indeed, if the points cluster near the true minimum, the deep model often achieves improvement in 1 or 2 iterations. With fewer points around the optimum the fit is poorer and more iterations are required to make an improvement. This variance in the number of steps before a better observation is acquired increases the variance of the regret.

We also examined the performance of residual deep GPs on Bayesian optimisation of the irregular target function without the initial shallow GP stage. We equip the last layer of the deep GP with the same hyper-priors as the shallow GP; however, instead of using the BFGS optimiser, we use L-BFGS due to memory constraints. Note that in the original experiment we did not use hyper-priors or a quasi-newton optimiser for the deep model. We use them here because we found that they are important for effective exploration with shallow GPs and they serve the same purpose for our deep GP.

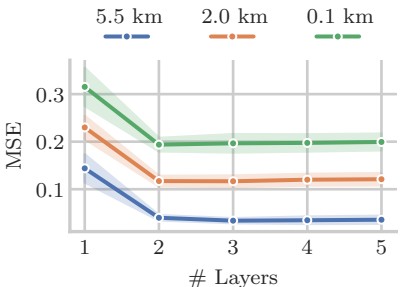

Figure 11: MSE of residual deep GPs on the wind modelling task across three altitude levels. Solid lines give the mean MSE, while the shaded regions around it span ±1 standard deviation. Statistics are computed from 12 runs—one for each month of 2010.

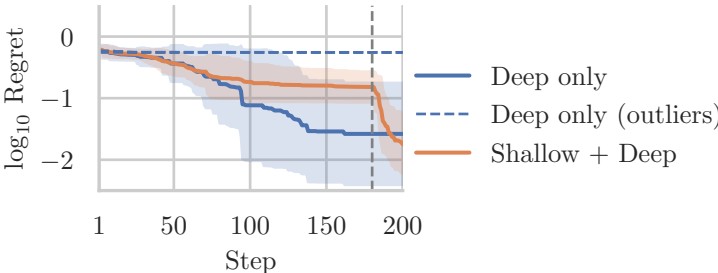

Figure 12: Comparison of Bayesian optimisation performed with a shallow GP followed by a residual deep GP vs only with residual deep GP on the irregular target function. Solid lines show median logarithm of regret while the shaded areas extend one standard deviation above and below. Blue dotted lines show three optimisation runs with the deep GP only which did not escape a local minimum—these runs contribute strongly to the large variance of its regret. Grey dotted line indicates a transition from the shallow GP to the deep GP at iteration 180.

We report the logarithm of regret achieved by using the residual deep GP model from the very first iteration in Figure 12. We find that in 12 out of 15 of the runs, our model improves upon the shallow GP, often even before the 100th iteration. This was expected, since we have seen that residual deep GPs recover shallow solutions when data is not abundant enough to capture target complexity. We also see that the variance of the regret is considerably larger than for the shallow GP. This is largely caused by the 3 outlier runs, indicated with a dotted blue line, where the model gets stuck in a local minimum. As this experiment is fairly sensitive to the setup setting, this could be due to the fact that the baseline is an exact GP, while our model recovers a sparse GP, and the fact that our model uses the L-BFGS optimiser, while the exact model uses the BFGS optimiser. Thus, using deep GPs exclusively can be more sample-efficient than employing the initial shallow GP stage; however, with the current setup it appears that this poses an increased risk of being stuck in a local minimum.

## A.5 WIND INTERPOLATION ON THE GLOBE

**Data**    To each location sampled along the track of the Aeolus satellite, we assign the wind velocity from the closest location present in the ERA5 dataset. Our test set is a grid of 5000 points, same as in Appendix A.3. To each location in the test set, we also assign the wind velocity from the closest location in the ERA5 dataset.

**Variational parameters**    For each GVF within the layers of the tested models we use 198 inter-domain inducing variables. They correspond to all vector spherical harmonics up to, and including, the 10th negative eigenvalue of the Hodge Laplacian. This choice is arbitrary and simply serves to balance quality of fit with training time.

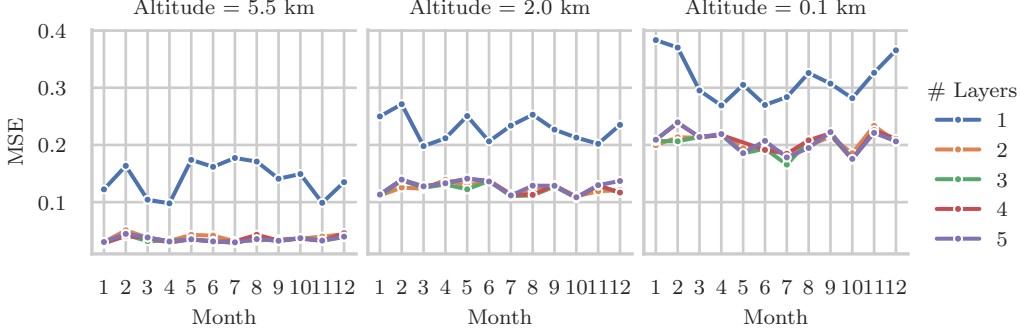

Figure 13: MSE of residual deep GPs on the wind modelling task across the 12 months of 2010.

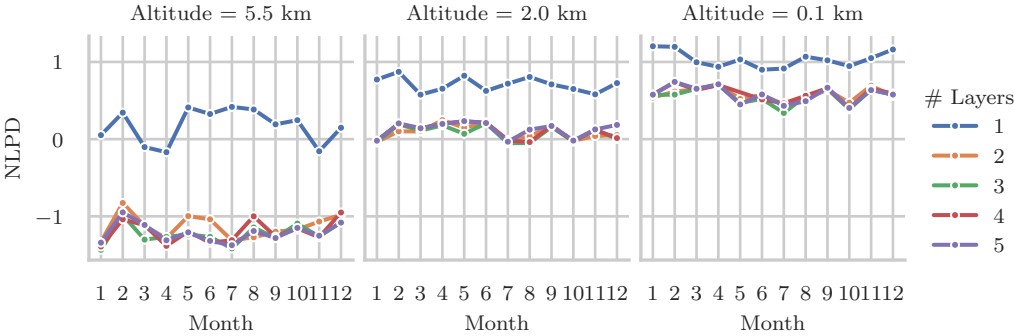

Figure 14: NLPD of residual deep GPs on the wind modelling task across the 12 months of 2010.

**Training and evaluation**   We fit the models to data using the Adam optimiser for 1000 iterations with the learning rate set to 0.01. To evaluate the models, we compute the MSE and NLPD via Monte Carlo sampling as described in Equation (16), we visualise the predictive uncertainty at a point $x_i$ which is computed as $\frac{1}{10} \sum_{n=1}^{10} \|\mathbf{S}_i^n\|$, where $\mathbf{S}_i^n$ is the posterior covariance matrix of the last layer given the $n$-th sample from the penultimate layer, and $\| \cdot \|$ is the Frobenius norm.

**Additional results**   The MSE comparison is presented in Figure 11. The individual results for each month are shown in Figure 14 and Figure 13. A larger version of Figure 5b as well as its analogs for other altitudes are presented in Figure 18. Figures 19 to 21 present the ground truth vector field, predictive mean and uncertainty, and one posterior sample for July 2010 for the three different altitudes.

Additionally, we compare our model with the baseline model from Section 4.1. The final layer of the baseline is a coordinate-frame GVF with independent Matérn 5/2 kernels, where the frame is given by the gradients of the spherical coordinates (with singularities taken care of), this is motivated in part by the approach of Mallasto and Feragen (2018). The results are shown in Figure 15 and Figure 16.

## A.6   REGRESSION ON UCI DATASETS

**Data**   In the mapping from $\mathbb{R}^d$ to $\mathbb{S}_d$, we set $b$ to 1. This is done for all datasets and the bias is kept constant during training. Indeed, in our initial examinations, we found that learning the bias often seemed to result in overfitting and worse performance.

**Training and evaluation**   We train each model for 5000 iterations using the Adam optimiser (Kingma and Ba, 2015) with the learning rate set to 0.01, such that the learning curves plateau

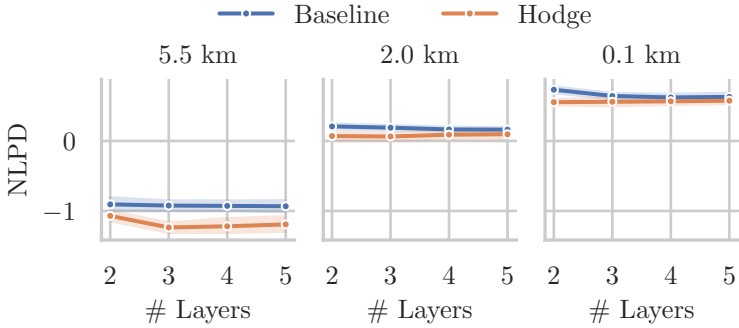

Figure 15: Comparison of the NLPD of residual deep GPs and the baseline model for wind field modelling across the 0.1 km, 1.0 km, and 5.0 km altitudes.

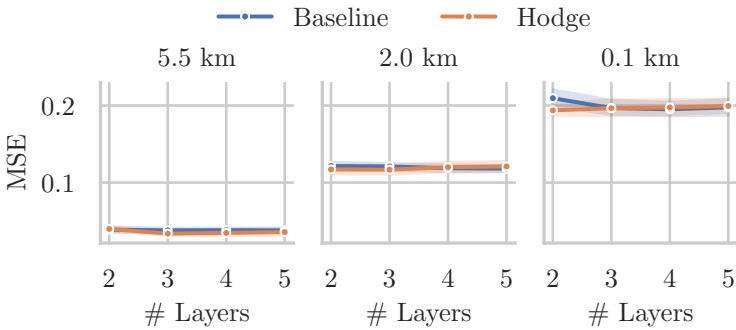

Figure 16: Comparison of the MSE of residual deep GPs and the baseline model for wind field modelling across the 0.1 km, 1.0 km, and 5.0 km altitudes.

for both Euclidean deep GPs and residual deep GPs. Each iteration consists of a gradient step using a batch of data. When the size of the training set is smaller than 1000 data points—that is, for the Yacht, Concrete, and Energy datasets—a batch is the entire dataset. For the Kin8mn and Power datasets, whose training sets are considerably larger, a batch of 1000 data points is sampled with replacement from the training set.

**Additional results** The test NLPD and MSE of all models can be seen in Figure 17.

## B   MORE ON INTERDOMAIN INDUCING VARIABLES ON MANIFOLDS

Empirically, we find that when the number of eigenfunctions $K$ used to approximate the manifold Matérn kernels exceeds the number of interdomain inducing variables, performance of residual deep GPs deteriorates. This can be surprising, since a higher $K$ yields a better approximation of the true manifold Matérn kernel.

A potential reason for this phenomenon may be identified by examining the kernel of a posterior sparse Matérn GP $f \sim \mathcal{GP}(\mu, k)$—more specifically, its whitened reparameterisation (see Appendix A.1), as that makes equations cleaner. In this, it will be helpful to define

$$\Psi_{i:j}(\cdot) = \left( \sqrt{\frac{\sigma^2}{C_{\nu,\kappa}} \Phi_{\nu,\kappa}(\lambda_i)} \phi_i(\cdot), \ldots, \sqrt{\frac{\sigma^2}{C_{\nu,\kappa}} \Phi_{\nu,\kappa}(\lambda_j)} \phi_j(\cdot) \right)^\top, \tag{37}$$

which allows us to express the Matérn kernel approximated with $K + 1$ eigenfunctions as $k_{\nu,\kappa,\sigma}(\cdot, \cdot') = \Psi_{0:K}(\cdot)^\top \Psi_{0:K}(\cdot')$. Now, recalling Equation (19), and denoting the number of inter-

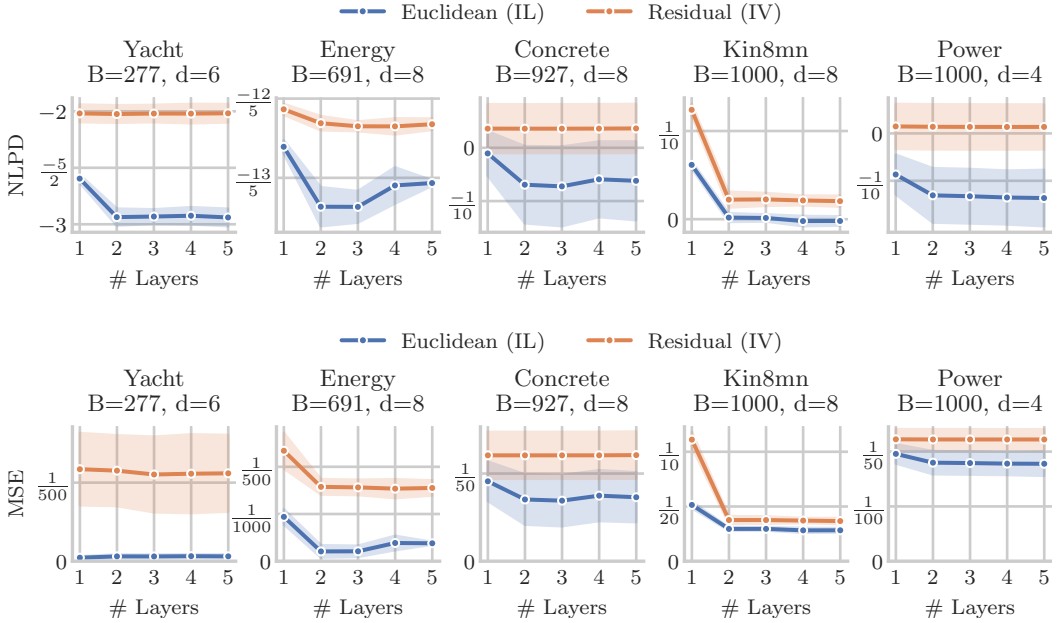

Figure 17: NLPD and MSE of residual deep GPs with spherical harmonic features and Euclidean deep GPs with inducing points on five UCI datasets. Residual deep GPs had their inputs mapped from $\mathbb{R}^d$ to $\mathbb{S}_d$ via $x \mapsto (x,b)/\|(x,b)\|$. Solid lines give the mean MSE and shaded regions around them span $\pm 1$ standard deviation. All statistics were computed from 5 randomised runs.

domain inducing variables by $M$, we see

$$k(\cdot, \boldsymbol{\zeta})\mathbf{L}^{-\top} = (\phi_0(\cdot), \ldots, \phi_M(\cdot))^\top \operatorname{diag}\left(\frac{1}{\frac{\sigma^2}{C_{\nu,\kappa}}\Phi_{\nu,\kappa}(\lambda_0)}, \ldots, \frac{1}{\frac{\sigma^2}{C_{\nu,\kappa}}\Phi_{\nu,\kappa}(\lambda_M)}\right)^{-1/2} \tag{38}$$

$$= (\phi_0(\cdot), \ldots, \phi_M(\cdot))^\top \operatorname{diag}\left(\sqrt{\frac{\sigma^2}{C_{\nu,\kappa}}\Phi_{\nu,\kappa}(\lambda_0)}, \ldots, \sqrt{\frac{\sigma^2}{C_{\nu,\kappa}}\Phi_{\nu,\kappa}(\lambda_M)}\right) \tag{39}$$

$$= \boldsymbol{\Psi}_{0:M}(\cdot)^\top, \tag{40}$$

which we can substitute into Equation (27)

$$k_{\boldsymbol{\zeta},\boldsymbol{m'},\mathbf{S'}}(\cdot, \cdot') = k(\cdot, \cdot') - k(\cdot, \boldsymbol{\zeta})\mathbf{L}^{-\top}(\mathbf{I} - \mathbf{S'})\mathbf{L}^{-1}k(\boldsymbol{\zeta}, \cdot'). \tag{41}$$

$$= \boldsymbol{\Psi}_{0:K}^\top(\cdot)\boldsymbol{\Psi}_{0:K}(\cdot') - \boldsymbol{\Psi}_{0:M}^\top(\cdot)(\mathbf{I} - \mathbf{S'})\boldsymbol{\Psi}_{0:M}(\cdot') \tag{42}$$

$$= \boldsymbol{\Psi}_{M+1:K}^\top(\cdot)\boldsymbol{\Psi}_{M+1:K}(\cdot') + \boldsymbol{\Psi}_{0:M}^\top(\cdot)\mathbf{S'}\boldsymbol{\Psi}_{0:M}(\cdot') \tag{43}$$

For $K = M$, posterior covariance reduces to the second term only, which is determined by the kernel hyperparameters and the variational covariance matrix. However, for $M > K$, the first term contributes additional variance that can only be reduced by changing the hyperparameters of the prior, like length scale and prior variance, rather than the variational parameters $\boldsymbol{m'}$ and $\mathbf{S'}$.

With this particular setup, there are two forces at play during optimisation: one which lowers the prior variance to match the posterior variance, and another which modifies $\mathbf{S'}$ to approximate the true posterior covariance by introducing dependencies between basis coefficients. In practice, we observe that this can lead to a difficulty if the $\boldsymbol{\Psi}_{0:M}^\top(\cdot)\mathbf{S'}\boldsymbol{\Psi}_{0:M}(\cdot')$ is already at the desired variance but $\mathbf{S'}$ must still be adjusted to approximate the true covariance, while $\boldsymbol{\Psi}_{M+1:K}^\top(\cdot)\boldsymbol{\Psi}_{M+1:K}(\cdot')$ is still too large. In this case, the first mechanism pushes $\boldsymbol{\Psi}_{M+1:K}^\top(\cdot)\boldsymbol{\Psi}_{M+1:K}(\cdot')$ downwards by lowering the prior variance $\sigma^2$, which necessarily also reduces $\boldsymbol{\Psi}_{0:M}^\top(\cdot)\mathbf{S'}\boldsymbol{\Psi}_{0:M}(\cdot')$. Consequently, $\mathbf{S'}$ must increase to compensate for this. This process results in a tug-of-war between the variational parameters and kernel hyperparameters, which seems to make optimisation difficult, and is thus a possible reason for the drop in performance as $K > M$.

One remedy is to set $K = M$, which is what we actually do in our experiments. However, since this comes at some cost to the kernel approximation, we propose that future work can consider an extended variational family which, in our preliminary tests, helped mitigate this issue at a minimal cost. Our extension expands the $\mathbf{S}'$ matrix with a parametrised diagonal $\mathbf{D}'$ corresponding to the $K - M$ eigenfunctions not previously used in the variational family but used in the kernel, giving

$$k_{\boldsymbol{\zeta}, \boldsymbol{m}', \mathbf{S}'}(\cdot, \cdot') \tag{44}$$

$$= \boldsymbol{\Psi}_{M+1:K}^\top(\cdot)\boldsymbol{\Psi}_{M+1:K}(\cdot') + \boldsymbol{\Psi}_{0:M}^\top(\cdot)\mathbf{S}'\boldsymbol{\Psi}_{0:M}(\cdot') + \boldsymbol{\Psi}_{M+1:K}^\top(\cdot)(\mathbf{D}' - \mathbf{I})\boldsymbol{\Psi}_{M+1:K}(\cdot') \tag{45}$$

$$= \boldsymbol{\Psi}_{0:M}^\top(\cdot)\mathbf{S}'\boldsymbol{\Psi}_{0:M}(\cdot') + \boldsymbol{\Psi}_{M+1:K}^\top(\cdot)\mathbf{D}'\boldsymbol{\Psi}_{M+1:K}(\cdot'). \tag{46}$$

This eliminates the aforementioned conflict, allowing the variational parameters to affect both terms in a similar way.

Ostensibly, when $K - M$ is large, it significantly impacts performance. However, as we have seen in Appendix A.1, this can be avoided by applying the addition theorem, which is possible if the parameters of $\mathbf{D}'$ corresponding to eigenfunctions with the same eigenvalues are kept equal. With this method, the number of additional variational parameters is minimal, and, comparing our extended variational family with its original variant, there is practically no increase in computation time, as $\boldsymbol{\Psi}_{M+1:K}^\top(\cdot)\mathbf{D}'\boldsymbol{\Psi}_{M+1:K}(\cdot')$ is simply substituted for $\boldsymbol{\Psi}_{M+1:K}^\top(\cdot)\boldsymbol{\Psi}_{M+1:K}(\cdot')$.

## C  RELATION TO WRAPPED GAUSSIAN PROCESSES

Wrapped GPs of Mallasto and Feragen (2018) are used to model a function of Euclidean data taking values on a Riemannian manifold. They are constructed by choosing a base point function, which assigns a point on the manifold to each Euclidean input point, and a coordinate-frame GVF prior (although the latter is implicit in the original paper). Inference is done by lifting the training labels from the manifold to the tangent space at the base points assigned to their corresponding inputs, performing inference in the tangent space, and projecting the posterior GVF into the manifold using the exponential map. The base point function is chosen either as a constant mapping to a point minimising the squared distance from the training points (i.e. empirical Fréchet mean) or as an auxiliary regression function.

We derived the manifold-to-manifold layers of our model as a generalisation of the linear mean construction of Salimbeni and Deisenroth (2017); however, we may also build them based on the ideas of wrapped GPs. The key difference is that our construction is manifold-input—whereas wrapped GPs are Euclidean-input—and the input and output manifolds are identical, allowing layers to be composed. Thus, the first non-trivial modification is to replace the Euclidean domain with the manifold domain. This requires adapting the GVF from Euclidean kernels to manifold kernels. Furthermore, instead of using an auxiliary regression function or Fréchet mean, the natural choice yielding a generalisation of the linear mean is the identity map for the base point function. This yields a manifold-to-manifold GP; however, to enable doubly stochastic variational inference, the next modification is to replace the exact GVFs with sparse GVFs using inducing points or interdomain inducing variables. With these modifications, we obtain our manifold-to-manifold GPs, which can be composed sequentially to yield residual deep GPs.

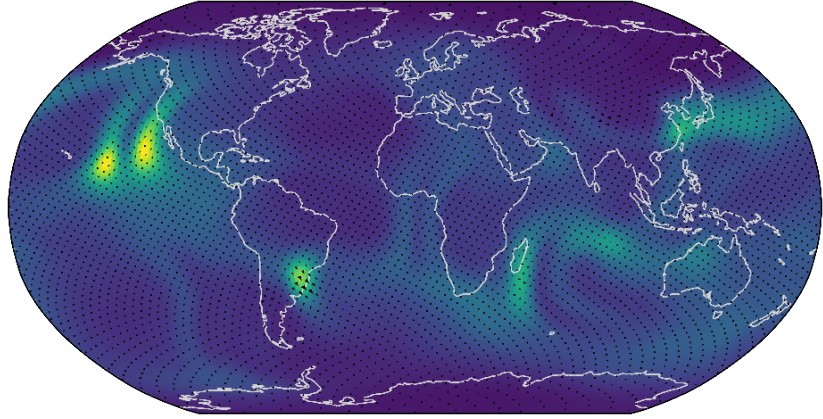

(a) 5.0 km altitude.

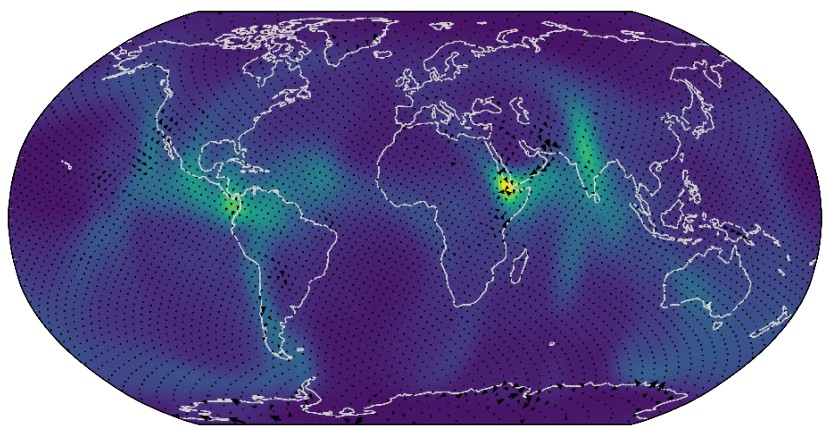

(b) 2.0 km altitude.

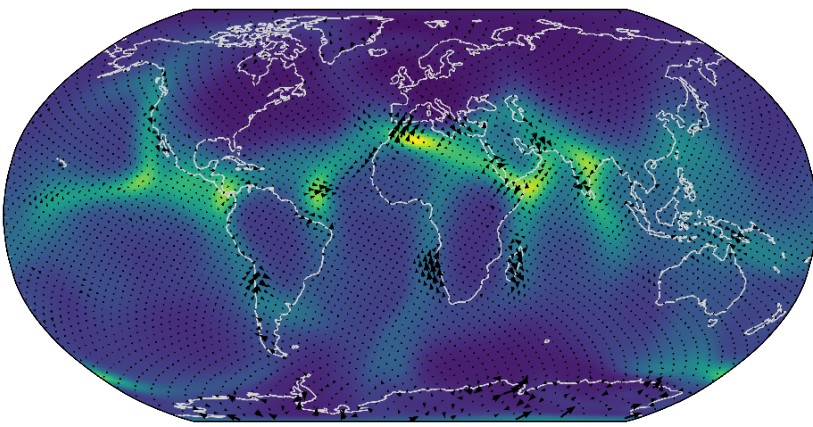

(c) 0.1 km altitude.

Figure 18: Difference between the prediction and the ground truth wind velocities, shown as black arrows, and the predictive uncertainty, shown using a colour scale from purple (lowest) to yellow (highest), for a 3-layer residual deep GP and wind velocities for July 2010, at three altitude levels.

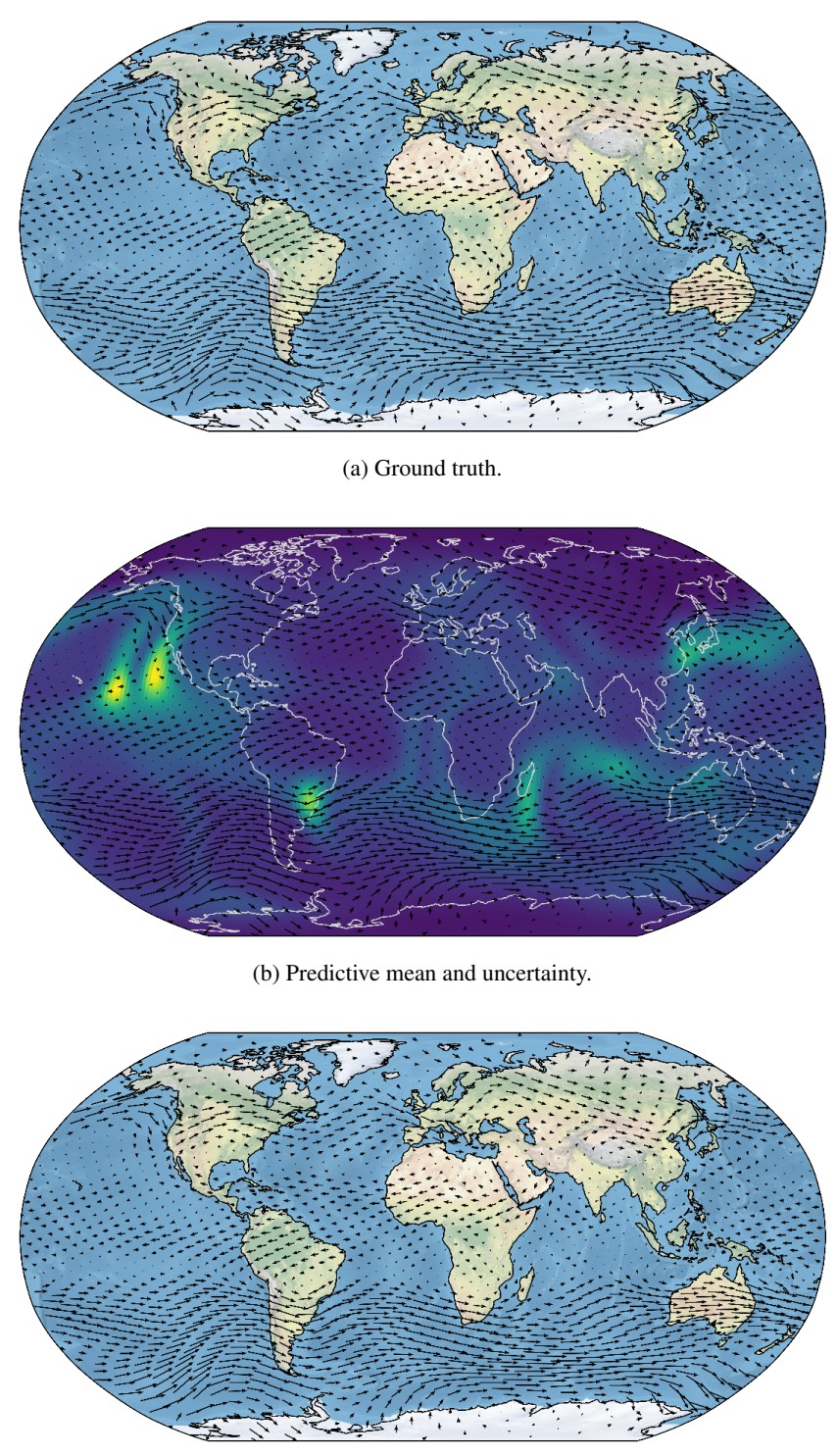

(a) Ground truth.

(b) Predictive mean and uncertainty.

(c) Posterior sample.

Figure 19: Ground truth wind velocity data at an altitude of 5.0 km from July 2010, and the corresponding posterior mean, uncertainty, and sample from a 3-layer residual deep GP. The mean and sample are shown as black arrows, while the predictive uncertainty is shown using a colour scale from purple (lowest) to yellow (highest).

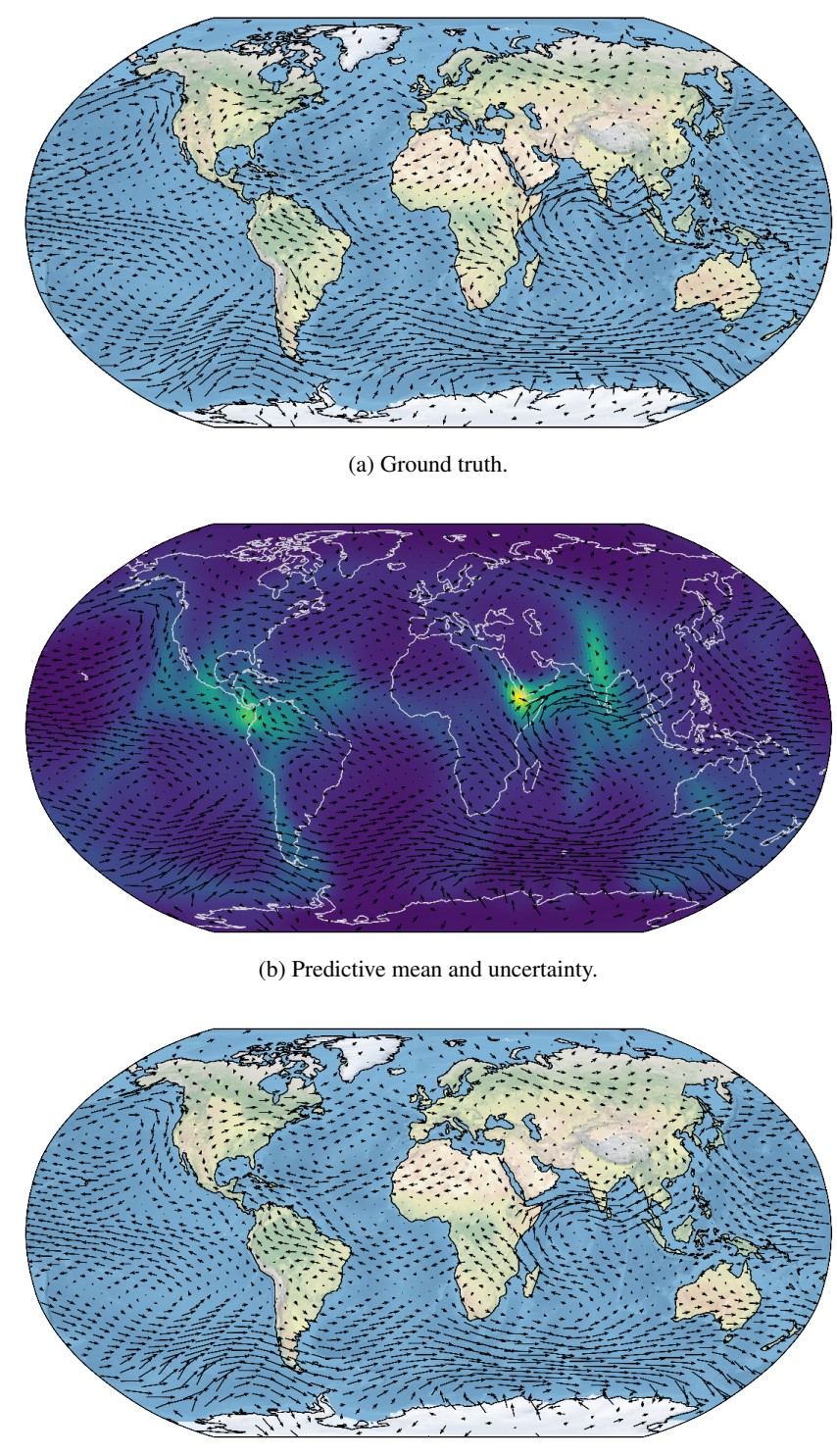

(a) Ground truth.

(b) Predictive mean and uncertainty.

(c) Posterior sample.

Figure 20: Ground truth wind velocity data at an altitude of 2.0 km from July 2010, and the corresponding posterior mean, uncertainty, and sample from a 3-layer residual deep GP. The mean and sample are shown as black arrows, while the predictive uncertainty is shown using a colour scale from purple (lowest) to yellow (highest).

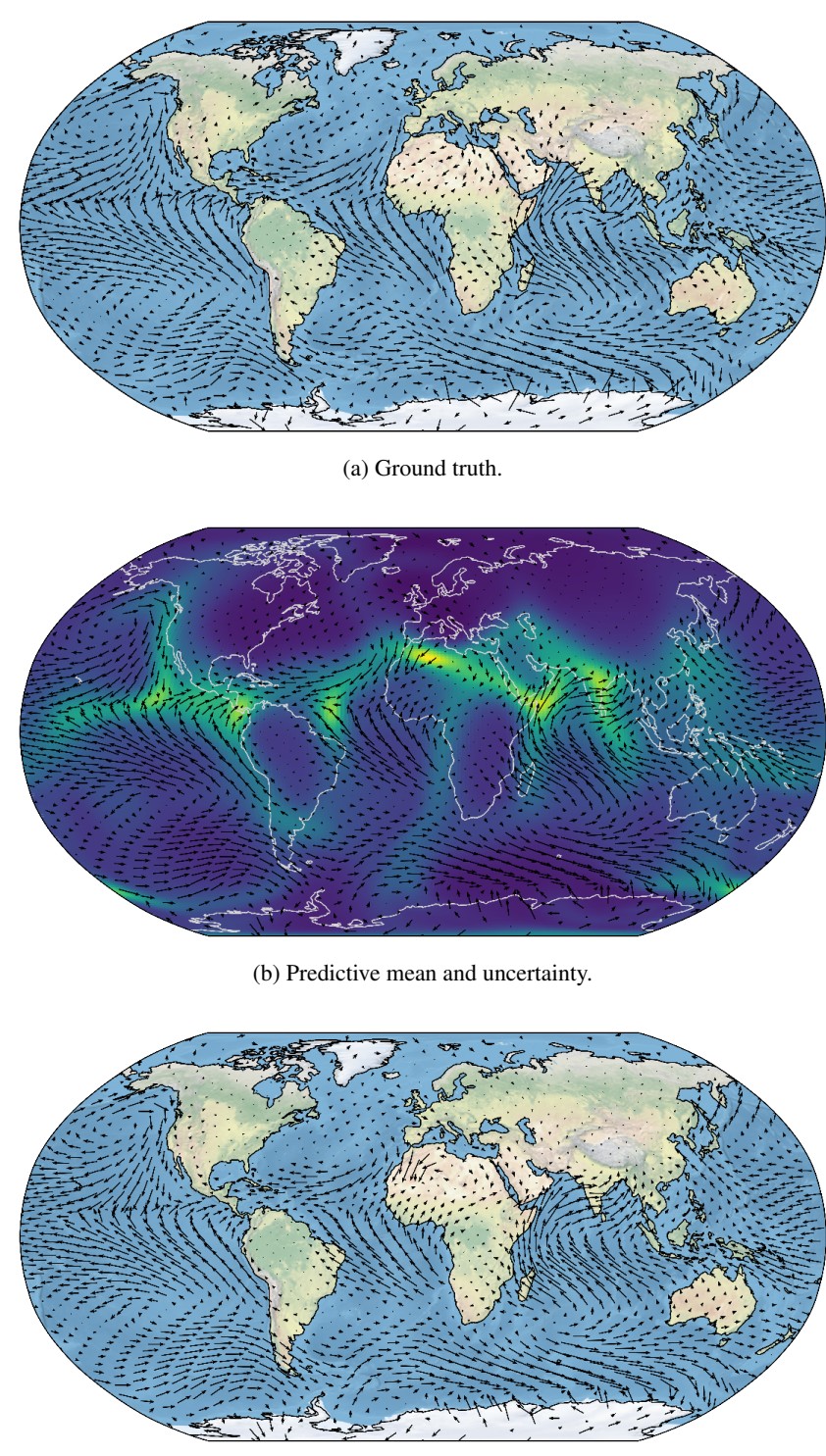

(a) Ground truth.

(b) Predictive mean and uncertainty.

(c) Posterior sample.

Figure 21: Ground truth wind velocity data at an altitude of 0.1 km from July 2010, and the corresponding posterior mean, uncertainty, and sample from a 3-layer residual deep GP. The mean and sample are shown as black arrows, while the predictive uncertainty is shown using a colour scale from purple (lowest) to yellow (highest).

