# OpenReview forum: "Residual Deep Gaussian Processes on Manifolds"
_ICLR.cc/2025/Conference — ICLR 2025 Oral_

### Official Review · Reviewer_ymXL · 2024-10-22

**Soundness:** 4
**Presentation:** 3
**Contribution:** 4
**Rating:** 8
**Confidence:** 4

**Summary:**

The authors proposed practical deep Gaussian process models on Riemannian manifolds by leveraging Gaussian vector fields and manifold-specific operations. Specifically, the proposed interdomain variables-based residual deep Gaussian processes show considerable improvement over the baseline for manifold data. Additionally, this approach can significantly accelerate inference (~2x) for Euclidean data while maintaining comparable performance. The claims made by the authors are well-supported by both rigorous theoretical development and extensive experimental validation.

**Strengths:**

* The paper is well-written and easy to follow

* The theoretical development is novel and solid

* The experiments are well-designed and thorough, starting from synthetic manifold data, progressing to real-world climate examples, with additional analyses of Euclidean data

* The appendix provides informative supplementary experiments that enhance understanding of the details

**Weaknesses:**

* The paper is technically dense and may be challenging for readers unfamiliar with Riemannian geometry. It would be helpful if the authors could provide a more detailed background on Gaussian processes in Riemannian geometry in the appendix. For practical use, I also encourage the authors to consider releasing the code for public use in the future

**Questions:**

See the weakness

---

> ### Author Response · Authors · 2024-11-23
> **Addressing Weaknesses**
>
> __Provision of background on GPs in Riemannian geometry. (Weakness 1)__
>
> Thank you for raising this issue.
>
> Indeed, it would be very helpful to provide the reader with an easy way to access a more complete introduction to GPs in Riemannian geometry.
> Fortunately, works that have initially developed these notions, and those that have later refined it, provide excellent introductions to the subject.
> Thus, instead of augmenting the appendix, we suggest that it is appropriate to instead direct the reader to these resources, which we now do in footnote 3.
>
>
> __Realising the code for public use.__
>
> Thank you for this suggestion.
>
> We plan to release the updated code base, with the additional experiments done for the discussion phase, shortly after the discussion closes.

---

> > ### Comment · Reviewer_ymXL · 2024-11-25
> >
> > Thank you for addressing my concerns. I believe this work represents an important step forward in the application of deep Gaussian Processes on manifolds

---

### Official Review · Reviewer_ezvQ · 2024-11-03

**Soundness:** 3
**Presentation:** 4
**Contribution:** 3
**Rating:** 8
**Confidence:** 4

**Summary:**

This work generalizes Deep Gaussian Processes to Manifolds. They address the key challenge of modeling each hidden layer being a manifold to manifold map by interpreting the output of each hidden layer as a Gaussian Vector Field combined with an exponential map, allowing each layer to learn a residual displacement on the manifold. From pre-existing works, they outline three different ways for constructing Gaussian vector fields (GVFs) on manifolds: Projected GVFs, Coordinate-frame-based GVFs and Hodge GVFs. They perform inference with this model by optimizing a variational objective.

**Strengths:**

1. I think this work is quite a natural and well-motivated extension of recent works on GPs on Manifolds. I find the perspective of defining hidden layer outputs as a residual(deviation from identity maps) to be quite ingenious since it leads to natural generalization to manifolds. As far as I know, this is a novel contribution.

2. I think the paper is quite well-written and very clear in its presentation.

**Weaknesses:**

My main concern is that I found the experimental results to be a bit mixed, some of the choices to be a bit confusing and the main claim of the paper(in 015 "Our models significantly improve performance in these settings ...")  to be not fully validated:
1. The choice of GVF across the experiments seems a bit ad-hoc. It's understandable that the Hodge GVF can only be used when the eigenfunctions of the Laplacian are known but 4.2 uses coordinate-based GVF and 4.4 uses projected gvf. This choice seems arbitrary and it would have been nice to see both the gvfs across all experiments.

2. Euclidean GPs are not considered as baselines in experiments 4.2 and 4.3. Additionally, in experiment 4.1 I am a bit concerned about the fairness of the comparison. It is stated that "we match the number of optimized parameters between models as closely as possible". I am not sure what this means and it would have been nice to make this concrete by having plots of wall-clock times for optimization of the different GPs or something similar. In experiment 4.4 euclidean GPs outperform the manifold variants by a margin. Additionally in experiment 4.1 while Figure 3 paints a flattering picture of the method proposed by the authors, figure 8 seems to paint quite a different picture. The baseline either matches (or outperforms?, It is hard to tell just from the image) the Hodge GVF in terms of MSE. Also, the trend looks like with additional layers deep Euclidean GPs would outperform all the manifold GP variants. As it stands I am not convinced that Residual Deep Gaussian Processes consistently outperform Euclidean Deep Gaussian Processes.

**Questions:**

See weaknesses

---

> ### Author Response · Authors · 2024-11-23
> **Addressing Weaknesses**
>
> __Motivation for GVF choice. (Weakness 1)__
>
> Thank you for bringing this issue to light.
>
> We used the projected GVF construction in Section 4.4, since we believe that it is generally more useful for practitioners, as it does not require choosing a coordinate frame.
> Despite that, wanting to showcase both constructions, we included results from coordinate-frame GVFs in Section 4.2, where we found little difference between models in our preliminary testing.
> We agree that our results would be more complete with both constructions in both experiments; however, we limited their scope to avoid overloading the presentation, especially since we did not want to emphasise the comparison between coordinate-frame and projected GVFs.
> Indeed, one of our key conclusions was to focus research on intrinsic approaches, such as the Hodge kernel, as they are superior not only in shallow GPs but also in deep models.
>
> __Parameter count matching. (Weakness 2)__
>
> Thank you for raising this issue.
>
> In experiment 4.1 we attempt to make the comparison fair by matching the number of trainable parameters between models.
> We give the precise numbers of variational parameters and kernel parameters per level in paragraph "Variational Parameters" in the Appendix section A.3, which we have now expanded to include precise formulae for improved clarity.
> Additionally, we ensured that if we cannot match the number of parameters exactly, the baseline model would be given the highest number, to prevent artificial handicapping for our model.
>
> __Baseline model with additional layers could outperform residual deep GPs. (Weakness 2)__
>
> Thank you for this very apt observation.
>
> To investigate the possibility of the baseline outperforming all manifold models, we extended experiment 4.1 for the baseline and Hodge models with up to 10 layers and doubled the number of randomised runs to 10 to increase confidence.
> We report the results in Figure 9 and Figure 10.
> We find that the baseline performance saturates more slowly giving the illusion that it might exceed that of the residual deep GP with deeper layers.
> However, what is especially clear with more data, it does saturate and never exceeds the performance of our model - neither in terms of NLPD nor MSE.
> In fact, with only three layers, our model surpasses the baseline across all depths.
>
> It is true that the MSE advantage of our model is smaller than its NLPD advantage.
> However, it is important to stress that NLPD assesses both the predictive uncertainty and the predictive mean, making it a more complete metric for GP evaluation than MSE.
> Indeed, if we were prioritising MSE, then we probably would not use GPs at all.
>
> __Euclidean baselines in Sections 4.2 and 4.3. (Weakness 2)__
>
> We thank the reviewer for bringing forth this issue.
>
> Firstly, however, it is important to mention that the results of these experiments are not in a vacuum with respect to Euclidean baselines, since they are both based on papers which show superior performance of shallow manifold GPs over Euclidean GPs.
> Nevertheless, we agree that a baseline of Euclidean deep GPs would bring additional context to Sections 4.2 and 4.3.
> Due to time restrictions we provided such a baseline only in experiment 4.3; however, we hope that the results will be of good help in assessing any advantage of residual deep GPs over Euclidean deep GPs overall.
>
> We report a comparison of our model with the baseline in terms of NLPD and MSE in Figures 14 and 15 respectively.
> We find that our model achieves a lower NLPD than the baseline for all altitudes and model depths.
> Furthermore, the lowest mean MSE is attained by the residual deep GP for every altitude level; however, for 4 and 5 layers at altitudes of 2.0 km and 0.1 km, and 2 layers at altitude of 5.0 km, our model falls very slightly short of the baseline.
> Thus, as in Section 4.1, while Euclidean models may attain similar MSE to residual deep GPs, our models provide better-calibrated uncertainty estimates, resulting in better NLPD performance.

---

> ### Author Response · Authors · 2024-12-02
> **Rebuttal Follow-up**
>
> Dear Reviewer ezvQ,
>
> Could you please confirm whether our rebuttal has adequately addressed your concerns?
>
> We are happy to provide any further details.

---

> > ### Comment · Reviewer_ezvQ · 2024-12-02
> >
> > Thank you for your response. It addresses most of my concerns. I raised my score accordingly.

---

### Official Review · Reviewer_GjkX · 2024-11-03

**Soundness:** 3
**Presentation:** 3
**Contribution:** 2
**Rating:** 6
**Confidence:** 3

**Summary:**

This paper outlines the idea of using compositions of Gaussian process priors on manifold structures. The basic idea is to take a normal Gaussian process with identity mean function, here referred to as a residual GP, and at each layer use the exponential map to constrain this to the tangent space of the manifold at x. This Riemanninan formulation has quite a nice connection to the identity mean function as the map itself is now defined by the location as in Eq 10. The paper then continues to fit the proposed modelling approach in how it fits into the literature of the doubly stochastic inference mechanism typically used for composite GPs and with interdomain inducing points to formulate the variational bound.

The paper concludes with a set of experiments, both synthetic experiments and a real experiment on a manifold.

**Strengths:**

This is a very well written paper, there are quite a few concepts to explain to get the background and the authors do this in a splendid manner. A lot of these topics are quite involved and it would have been easy to stray of the necessary path but the authors do an excellent job of not doing this. Very concrete and to the point, well done!

The derivation of the model is well done keeping a good balance between details and intuition of the approach.

**Weaknesses:**

A substantive assessment of the weaknesses of the paper. Focus on constructive and actionable insights on how the work could improve towards its stated goals. Be specific, avoid generic remarks. For example, if you believe the contribution lacks novelty, provide references and an explanation as evidence; if you believe experiments are insufficient, explain why and exactly what is missing, etc.

If I should play devils advocate with this paper the authors do not do enough to highlight the novelty of their approach neither highlight the benefits except for experimentally. We know that composite GPs are horrific to train, while the doubly stochastic bound often allows us to train the model it often leads to very questionable uncertainties. The authors do not do enough to highlight how, or if, this approach addresses these issues. These might come as very generic comments as it more relates to composite GPs in general but this is also how the authors mainly "sell" the paper, that it is a benefit in general to have a manifold structure even if I do not know the manifold of the actual data. It would be great to hear the authors address this.

I also find the Bayes Opt experiment a little bit disappointing, the idea of switching model after a specific set of iterations makes sense but how do I know where to switch, is this the optimal choice that is shown and if so what happens in other locations, both earlier and later? It would be good to see a more in-depth comparison here, what is the result if I use the composite GP from iteration 1? What happens if I swap to another shallow model at the same time? How are the hyper-parameters dealt with, do you marginalize them at each iteration? What is also worrying is the variance in the composite GP doe seem to go up quite substantially even while having 175 points to start from. It would be great to see a few different functions not just Ackley to see if the behavior is consistent across experiments.

**Questions:**

- While the work is theoretically sound I would love to see more analysis of why it works well. My conclusions from the paper are two, the first, if my data lies on a manifold then of course I should constrain my prior to reflect this. However, it does also seem to be the case that this approach performs better in general. I would love to try and get an intuition into this. We know after 10 years of working with composite GPs that simply they do not work, there are too many symmetries in the structure and together with the lax approximations this makes them very hard to use. What of this does this approach address? I would love to get some insights into this from the authors.
- In general it would be great to have more explanation of the model set-up for the different experiment, what are the thoughts behind specific model choices etc. As far as I understand from the experiment you perform modelling on the hyper-sphere, why is this a motivated choice?
- I would love to see an experiment where you actually have known manifold structures that are combined in a composition. What would be interesting to see is if you can correctly factorise the uncertainty across the different layers in the composition.
- It would be great if the authors could make more comments related to paper [1] which is at the moment only referenced as a footnote comment.
- Would be great if you could expand the caption to Figure 2 to include explanations to both a and b. Right now there is a lot of jumping between pages to get that caption.

[1] Mallasto, A., & Feragen, A. (2018). Wrapped gaussian process regression on riemannian manifolds. In , 2018 IEEE/CVF Conference on Computer Vision and Pattern Recognition (pp. 5580–5588). : .

---

> ### Author Response · Authors · 2024-11-23
> **Addressing Questions and Weaknesses (Part 1)**
>
> __How to choose when to switch the models and is the 180th iteration optimal? (Weakness 2)__
>
> Thank you for this question.
>
> With the switching strategy, we aimed to show that by only using a deep GP when the shallow GP gets stuck, we can achieve large performance improvements without a runtime penalty.
> Indeed, the choice of the 180th optimisation step is not optimal in terms of attaining the lowest regret at step 200 - instead, it was chosen so that it is evident from the flat regret curve that the shallow GP is stuck by the time we switch to a deep model.
> Choosing when to switch is non-trivial and largely depends on the compute budget, since, as seen below, we can often achieve better results by employing only deep GPs.
> What our results do suggest is that an adaptive strategy, such as switching periodically to a deep GP when the shallow GP is stuck, could achieve a good balance of compute to performance.
>
> __Using composite GPs from iteration 1. (Weakness 2)__
>
> Thank you very much for this suggestion.
>
> Employing residual deep GPs from the very beginning is certainly possible, since at low data densities the model recovers a shallow GP.
> The only difference is that it would recover a sparse shallow GP, whereas in Section 4.2 we use an exact shallow GP.
> To see the actual behaviour, we performed 15 deep-GP-only BO runs with the irregular target function.
> Through this process we found that the combination of hyper-priors and quasi-Newton optimisation employed in Jaquier et al. (2022) is useful for effective exploration with the shallow GP and, indeed, with the deep GP too (we remark about this behaviour in Appendix A.4).
>
> We report the results of this experiment in Figure 12.
> In 12 runs, the deep GP found a point with significantly lower regret than shallow GPs - often in fewer than 100 iterations.
> In 3 runs, however, the deep GP got stuck in a local minimum.
> The reasons for this are not fully clear.
> It may be caused by the approximate nature of the shallow GP recovered by the deep GP or by using the L-BFGS optimiser instead of BFGS as for the exact shallow GP - regardless, additional experiments would be needed to explain this behaviour.
> Despite this, if sample-efficiency is key, Bayesian optimisation with deep GP from the start can be more effective than the synthetic strategy beginning with a shallow GP.
>
> __Switching to a different shallow GP. (Weakness 2)__
>
> Thank you for this question.
>
> To find out, we performed 10 randomised BO runs of the irregular function, switching to a shallow GP with the manifold squared-exponential kernel at iteration 180.
> We found no improvement in the quality of minima found, despite the change of kernel.
>
> __Hyper-parameters and hyper-priors. (Weakness 2)__
>
> Thank you for this question.
>
> The hyper-priors, which we set following Jaquier et al. (2022), are only used in the training of the models and do not play a role at inference time.
> In particular, we do not marginalise the hyper-parameters.
>
> __Increased variance of regret after switching to a deep model. (Weakness 2)__
>
> Thank you for this astute observation.
>
> Fortunately, the reasons for this increase in the variance of regret after switching to residual deep GPs can be easily elucidated, which we now do in the paragraph "Additional results and analysis" of Appendix A.4.
> This increase in variance is due to the fact that there is a large variation in the initial 180 points acquired by the shallow GP between different runs.
> The locations of these points influence the quality of fit of the deep GP and, if the points were particularly close and dense around the optimum, the deep model often improves after 1 or 2 iterations, but with fewer points around the optimum we find the fit poorer and the deep model requires a higher number of iterations and, although rarely, sometimes 20 is not enough.
> This variance in the number of steps needed after the transition to find a better optimum leads to the increased variance in regret.
>
> __Additional benchmark functions. (Weakness 2)__
>
> Thank you for this suggestion.
>
> We agree that additional benchmark functions would help examine the behaviour of Bayesian optimisation with our model more broadly.
> In fact, it would be very interesting to dive deeper into geometry-aware Bayesian optimisation with residual deep GPs, the different strategies for their use with shallow GPs, and their use in various optimisation landscapes; however, we believe that a more thorough exploration is better suited for a follow-up work.
> Thus, we decided to limit the number of examined functions, balancing the sizes of different experiments for a more well-rounded examination of our model.

---

> > ### Comment · Reviewer_GjkX · 2024-11-25
> > **Thank you for the comments**
> >
> > Thank you for addressing my questions and providing clarifications on the work. I am happy that my initial score of the paper reflects the work well.

---

> ### Author Response · Authors · 2024-11-23
> **Addressing Questions and Weaknesses (Part 2)**
>
> __Benefits of residual deep GPs without a known manifold. (Question 1)__
>
> Thank you for this question.
>
> Our results do show that residual deep GPs are effective models for manifold-input data, outperforming shallow manifold GPs and Euclidean deep GPs with a geometry-aware input layer.
> Moreover, we do suggest that, if an auxiliary manifold that does not degrade prediction quality is found, our model could serve as a faster alternative to Euclidean deep GPs.
> However, we do not intend to imply that our model ought to be used where the manifold of the data is unknown---as kindly remarked by reviewer a17f, we deliberately make this limitation known in Section 4.4.
> If there is any part of the text that is misleading, we are eager to correct it.
>
> __Overcoming training issues and questionable uncertainty estimates. (Question 1)__
>
> Thank you for this question.
>
> Our results suggest that training and uncertainty estimation are not an issue---at least insofar as they are not an issue for shallow GPs.
> Indeed, all experiments show that our models outperform shallow GPs in MSE and, most importantly, NLPD.
> In particular, in Section 4.1 we find that initialising the variance of GVFs to a small value, e.g. 0.0001, along with the residual structure, helps the model recover a shallow solution where data is scarce.
> Additionally, in Section 4.3, we show that this improvement in metrics translates to an improvement in visual interpretability of predictions.
> Indeed, we see that predictive uncertainty of residual deep GPs captures information not only about how far a test point is from a training point (as in the case of shallow GPs) but also about the behaviour of the nearby training values---e.g., regions with large differences between nearby wind velocities have high predictive uncertainty.
>
> __Motivation for modelling on the hyper-sphere. (Question 2)__
>
> Thank you for this question.
>
> The choice of hypersphere as the auxiliary manifold in Section 4.4 is motivated by the work of Dutordoir et al. (2020).
> To justify this choice, they note that GPs representing limits of neural networks have kernels which normalise their inputs, and thus can be seen as kernels on hyperspheres.
>
> We focus on hyperspheres in our work as a whole because: 1. they are spaces of practical relevance to many real-world problems; 2. they make the application of manifold-specific techniques quite simple.
> These two advantages were crucial for us, since our goal was to evaluate the real-world relevance of our model and examine how it fits in with existing manifold GP machinery.
> Nevertheless, our choice to focus on hyperspheres does not limit the applicability of our model to other manifolds.
>
> __Combining known manifolds in composition. (Question 3)__
>
> Thank you for this interesting suggestion.
>
> Indeed, if we understand correctly, it is suggested to use different manifolds at subsequent layers.
> This is a very interesting problem---one that we have thought about, but is beyond the scope of our current paper.
> One can immediately imagine constructing such sequences of manifolds through inclusion, projection, or other more complicated maps; however, we believe this would require extensive exploration and additional theoretical consideration better suited for a follow-up work.
>
> __Relation of residual deep GPs to wrapped GPs. (Question 4)__
>
> Thank you for suggesting that this topic should be expanded upon.
> We hope to have addressed your question in the general response.
>
> __Expand caption to Figure 2. (Question 5)__
>
> Thank you for this suggestion.
>
> Adding an in-caption explanation of the GVF constructions would certainly help avoid jumping between pages.
> We will attempt to make space for and expand the caption once all other changes are finalised at the end of the discussion process.

---

### Official Review · Reviewer_VHwa · 2024-11-03

**Soundness:** 4
**Presentation:** 4
**Contribution:** 3
**Rating:** 8
**Confidence:** 2

**Summary:**

This paper proposes a manifold-to-manifold deep GP inspired by residual neural networks. It only requires modifying the last layer with a Gaussian vector field that represents the residual from the identity map. The input to that layer is then translated by the residual using an exponential map. The authors demonstrate that this model is effective on a wide range of synthetic and real-world experiments.

Disclaimer: While I have expertise in GPs, I don’t know much about non-Euclidean spaces or manifolds except for a high level understanding. Therefore, I might underestimate or overestimate the technical contributions of this paper. This is reflected in my low confidence score.

**Strengths:**

1. This paper proposes a simple and effective way to construct manifold-to-manifold deep GPs by modeling the residuals, which is both novel and interesting.

2. The paper is well-written with a good logic flow. It provides a clear introduction of the relevant technical background and a discussion of related works, which makes it easy for readers to understand the context of this work. The description and derivation of the model is also clear and easy to follow. The beautiful figures are a nice addition to the excellent content of the paper.

3. The proposed model is thoroughly evaluated on a wide range of synthetic and real-world experiments, including a careful ablation study of different choices of Gaussian vector fields and variational families. Empirical results show that the model consistently provides improved predictive accuracy and robust uncertainty estimation compared to shallow manifold GP baselines, which demonstrates its usefulness and wide applicability.

**Weaknesses:**

It seems that the deep GPs saturate quite quickly as depth increases in practice as shown in Figures 3 and 6, especially when the number of data points is small (which is the setting where GPs are typically employed). More generally, deep GPs can be difficult to train and not very scalable in practice, due to, e.g., mode collapse in variational inference. I guess residual deep GPs also share these limitations or perhaps even worse due to the additional components for handling manifolds. In the conclusion section, could the authors comment more on the limitations of residual deep GP (e.g., balance between dataset size, effective depth, scalability, accuracy, uncertainty quality, robustness, approximation, etc) and provide some suggestions as to when residual deep GPs should be preferred over, say, manifold-to-manifold neural networks (and potentially their ensembles if uncertainty estimation is required)?

**Questions:**

See weaknesses.

---

> ### Author Response · Authors · 2024-11-23
> **Addressing Weaknesses**
>
> __Additional details on limitations of residual deep GPs__
>
> Thank you for this excellent suggestion.
>
> It is true that in most of our experiments residual deep GPs saturate quickly with increasing depth, especially when data is scarce.
> We should stress, however, that this accords with the findings of Salimbeni and Deisenroth (2017) for Euclidean deep GPs, who additionally show that deeper layers become useful with much more data (up to a billion data points).
> We find an analogous trend of reduced saturation with increased data density in Figure 3; however, it would be useful to investigate whether this pattern holds for much larger real world datasets.
> We now highlight this limitation as a potential research direction in the Conclusion section.
>
> __When to choose residual deep GPs over manifold neural networks?__
>
> Thank you for this question.
>
> Unfortunately, the literature on manifold neural networks is scarce and practically non-existent for their probabilistic formulations.
> Thus, any judgement of preference for one over the other we would make would be chiefly speculative.
> A comparison between these two frameworks would be certainly valuable;
> however, due to the additional work necessary on the side of neural networks, we believe it is better suited for future research.

---

> ### Comment · Reviewer_VHwa · 2024-11-25
>
> Thank you for your responses. Regarding the second point, one simple practical version of probabilistic manifold neural networks would be using deep ensembles, but I agree with the authors that additional work on the manifold neural network side would be necessary for a meaningful comparison.
>
> I would encourage the authors to include the discussions in their rebuttal regarding limitations of residual deep GPs and perhaps the future research of manifold neural networks in the revision of the paper.

---

### Official Review · Reviewer_a17f · 2024-11-03

**Soundness:** 3
**Presentation:** 4
**Contribution:** 3
**Rating:** 8
**Confidence:** 3

**Summary:**

The authors propose a technique for manifold->manifold GP regression, which they propose to use as a component for deep GP regression on manifolds. They illustrate their procedure via a battery of numerical experiments on real and synthetic problems, including regression and Bayesian optimization.

**Strengths:**

---

Excellent presentation

Between the excellent illustrative figures such as figs 1/2 and the compelling narrative flow, this paper is a pleasure to read, which is especially outstanding given the proclivity of some authors to lean into the formalism when discussing ML on manifolds.

---

Clear and consequential applications

The authors make a good case that Deep GPs can be superior to single-layer GPs, and that manifold data are important, such that getting deep GPs for manifolds is an important problem to solve.

**Weaknesses:**

Since this paper uses numerical experiments rather than theory to introduce its methodology, the experiments are crucial to get right. I have some comments on them below.

---

Need clearer comparison to prior manifold->manifold GPs.

It would be help clarify the novelty and contribution of this article to make a more direct comparison to prior work with GPs on manifolds. Probably it is obvious to the authors, but it would be helpful to say exactly what the shortcomings of Mallasto and Feragen's 2018 work, say, which this work addresses. It would also be interesting to compare this methodology to theirs, or to explain why this is not feasible. See Questions for more.

---

Regarding BO on Ackley.
"This outcome aligns with our expectations, as the region around the minimum of the Ackley function is smooth and can be effectively modelled by a shallow GP."
But the model is modeling the entire function, not just a region around the optimum, right? And the ackley function exhibits significant nonstationarity, so we would expect a deep GP to do a better job modeling it. How do the two methods compare in terms of out-of-sample MSE on this function? Does the deepGP do a better job modeling it, but just not optimizing it?

---

 Section 4.4

I'd like to first commend the authors for sharing a negative result; the failure to investigate limitations of methods is doubtless a serious drag on machine learning research.
However, I have methodological concerns about this study, such that I fail to be convinced that a negative result is indeed appropriate.
i) The per-iteration complexity is interesting, but it's not the ultimate issue of concern, which is wall-time to a given accuracy (as iterations required may differ between methods).
ii) The use of different optimization methods makes the comparison difficult; I understand the idea may have been to make a comparison to the existing baseline rather than a different one, but it's crucial to identify where any advantage or lake thereof lies by using the Euclidean GP with Adam as well.
iii) It would be great if you could clarify on where the memory requirements for L-BFGS are coming from; a priori, one would expect that, since Adam keeps track of two vectors of the same size as the parameter vector, using L-BFGS with a history of size 2 would have similar memory requirements, and it seems like it would be possible to use a slightly larger one, like size 10.

**Questions:**

1) Can you please make more clear for readers without expertise in differential geometry how your approach to defining a manifold-valued stochastic process differs from that of Mallasto and Feragen, who also use an exponential map on the tangent space at a given point? Please emphasize the practical implications of the differences.
Also, is it the case that that method could not support a deep GP, or that there would be considerable work required to make a deep GP out of it?

2) Does your work have implications for spaces that have much more structure than a manifold, like vector or function-valued mappings? Or do we recover already-used methods when deploying it in such context.

---

> ### Author Response · Authors · 2024-11-23
> **Addressing Questions and Weaknesses (Part 1)**
>
> __Comparison to wrapped GP regression. (Weakness 1 and Question 1)__
>
> Thank you for suggesting that this matter be discussed in more detail.
> We hope to have answered your question in the general response.
>
>
> __Why do residual deep GPs not improve upon shallow GPs in BO on Ackley. (Weakness 2)__
>
> Thank you for bringing this point to our attention.
>
> Indeed, because the Ackley function exhibits significant non-stationarity, residual deep GPs do model it better in regions where enough data is present to capture the function's complexity.
> In particular, with a grid of points on the sphere, the residual deep GP we used models the entire function better than the shallow GP, both in terms of MSE and NLPD as seen in the Table below for 100, 200, and 400 points.
> As in the original experiment, we set the inducing locations to the training data locations.
>
> |                     | Deep  $(N=100)$ | Shallow    $(N=100)$ | Deep     $(N=200)$ | Shallow     $(N=200)$     | Deep    $(N=400)$         | Shallow   $(N=400)$       |
> |---------------------|------------------|------------------|------------------|------------------|------------------|------------------|
> | **NLPD**          | 0.217           | 0.681           | 0.128           | 0.830           | -0.218          | 0.975           |
> | **MSE**            | 0.082           | 0.091           | 0.067           | 0.072           | 0.031           | 0.050           |
>
>
> Yet, in spite of these results, the lack of improvement of the deep model over the shallow GP in Bayesian optimisation is not an anomaly.
> This is because typically at iteration 180 the shallow model has already found the true minimum mode; thus, to make any improvements, it is only necessary to model the function in a region around that mode.
> Since in that region the function is smooth, the shallow GP is as effective as the deep GP at modelling it.
> We designed the irregular function to showcase what can happen when a minimum is not in a smooth region but rather is approached by nearing a singularity of the objective function.
> In this case, when the models are switched the shallow model has also typically found the mode but is unable to accurately model the region near it due to the singularity---here the deep GP can yield an improvement.
>
> We hope that the modified phrasing of the last sentence in the "Results" paragraph of Section 4.2 conveys the explanation above.
>
> __Optimiser choice. (Weakness 3)__
>
> Thank you for bringing this issue to light.
>
> Firstly, it is important to stress that both our model and the Euclidean baseline are trained using the Adam optimiser, making the comparison between them valid in its own right.
>
> Nevertheless, our choice of optimiser is not arbitrary.
> Chiefly, we choose Adam as it alleviates memory issues via mini-batching, while the L-BFGS implementation in large libraries such as Scipy or Optax are classical and do not support stochastic optimisation---unlike some of the recent L-BFGS variants (Byrd et al. 2014, Moritz et al. 2016).
> Thus, Adam is more convenient both for us and for anyone drawing on our results in the future.
> Secondly, Adam is used by Salimbeni and Deisenroth (2017), making our results more readily comparable with their experiments, which gives useful context to our results, since residual deep GPs generalise their linear mean model.
> It is true that matching our optimiser with Salimbeni and Deisenroth (2017) causes a mismatch with the work of Dutordoir et al. (2020).
> However, we believe that the preceding advantages to outweigh this disadvantage, especially since, even with L-BFGS, we would not be able to replicate the experimental setup of Dutordoir et al. (2020) because the collapsed ELBO objective they use is intractable for deep GPs.
>
> __Wall-clock time to a given accuracy. (Weakness 3)__
>
> Thank you for this suggestion.
>
> We believe that, if our model could achieve the accuracy of the Euclidean model, a wall-clock time to accuracy metric would be most appropriate.
> However, we chose the number of training iterations so that both models approximately reach convergence; thus, in this setting residual deep GPs cannot achieve the performance of Euclidean deep GPs---no matter the wall-clock time.
> For this reason we choose to report per-iteration wall time and performance separately.
> Indeed, although we report a negative result, with this separation we hope to make clear that, if the performance improved for a different (compact) auxiliary manifold, we could have an alternative to Euclidean deep GPs that is about twice as fast.
>
> We have clarified the reason behind the number of training iterations in Appendix A.

---

> ### Author Response · Authors · 2024-11-23
> **Addressing Questions and Weaknesses (Part 2)**
>
> __Implications for vector- and function-valued mappings. (Question 2)__
>
> Thank you for this interesting question.
>
> For finite-dimensional vector spaces, we have an answer.
> Since such spaces are isomorphic to the Euclidean manifold of the same dimension, our model reduces to the linear mean deep GP of Salimbeni and Deisenroth (2017).
> For function spaces which are compact or non-compact symmetric Riemannian manifolds we can apply our model directly, though non-compact manifolds, e.g. that of positive semi-definite bilinear forms, require sampling from the spectral density of the kernel when approximating its series expression (Azangulov and Smolensky et al. 2023).
> For infinite-dimensional function spaces, this might be an interesting research direction.

---

> > ### Comment · Reviewer_a17f · 2024-11-26
> >
> > Thanks for your reply; that helps. I see: the scalability issue with BFGS is not storing the preconditioner but just forming the gradient without minibatching.
> >
> > Anyways I've reread the paper, your reply, and my fellow reviewer's responses and decided to increase my score.

---

### Author Response · Authors · 2024-11-23
**General Response**

We thank all reviewers for their thorough analysis, excellent questions, and valuable suggestions.
A request for additional background on manifold-valued GPs and their relation to our work, especially in the context of wrapped GPs (Melasto and Feragen, 2018), appeared in two of the reviews.
It is certainly a beneficial topic to discuss, which is why we have incorporated a version of the note below in Appendix C, referenced in the main text.

Melasto and Feragen (2018) introduce a method (Algorithm 4 in the paper) of performing GP regression given Euclidean inputs and manifold outputs.
In short, to perform wrapped GP regression, one chooses a base point function, which assigns a point on the manifold to each Euclidean input point, and a coordinate-frame GVF prior.
Then, one lifts the training labels from the manifold to the tangent space at the base point assigned to their corresponding inputs using the logarithm map, performs inference in the tangent space, and projects the posterior GVF into the manifold using the exponential map.
The base point function is chosen either as a constant mapping to a point minimising the squared distance from the training points (i.e. empirical Fréchet mean) or as an auxiliary regression function.

Although our manifold-to-manifold layers were derived as a generalisation of the linear mean construction of Salimbeni and Deisenroth (2017), they can also be viewed as a modification of wrapped GPs.
The key difference is, of course, that the domain of the layers in our model is a manifold.
Moreover, the input and output manifolds are identical, allowing these layers to be composed.
Since wrapped GPs work with Euclidean data, to match our construction, we must make the non-trivial modification of the domain to be the output manifold.
This requires that, instead of using an auxiliary regression function or Fréchet mean, we use the identity map for the base point function.
Moreover, since the input data is now on a manifold, we must adjust the kernels accordingly; thus, instead of Euclidean kernels we use appropriate manifold kernels - in particular, the Hodge kernel, enabling intrinsic GVFs.
Finally, to enable doubly stochastic variational inference, we replace exact GVFs with sparse GVFs using inducing points or interdomain inducing variables.
After such modifications, we obtain manifold-to-manifold GPs, which can be composed sequentially to yield a residual deep GP.

---

### Author Response · Authors · 2024-12-04
**Thank you**

We thank the reviewers for this fruitful discussion period. Your thorough and insightful feedback has helped us meaningfully improve our work, and we are grateful for the thoughtful consideration you gave to our rebuttal. Once again, we sincerely appreciate the time, effort, and care you dedicated to this process.

---

### Meta-Review · Area_Chair_xeek · 2024-12-19

**Metareview:**

The paper 'Residual Deep Gaussian Processes on Manifolds' was reviewed by 5 reviewers who gave it an average score of 7.6 (final scores: 6+8+8+8+8). The reviewers appreciated the presentation, even if they also pointed out that it is rather dense and technical. The experiments were found a bit mixed, but still interesting. All reviewers argue for accepting the paper.

As recommended by reviewer VHwa, it would be good to include the requested discussion in the camera-ready paper.

**Additional Comments On Reviewer Discussion:**

This paper had many reviewers, all five of whom were active in the discussion phase. The average score increased from 6.8 -> 7.6 during the discussion.

---

### Decision · Program_Chairs · 2025-01-22

Accept (Oral)